# The control of tonic pain by active relief learning

**Suyi Zhang**[1,2]*, **Hiroaki Mano**[1,2,3], **Michael Lee**[4], **Wako Yoshida**[2], **Mitsuo Kawato**[2], **Trevor W Robbins**[5], **Ben Seymour**[1,2,3]*

[1]Computational and Biological Learning Laboratory, Department of Engineering, University of Cambridge, Cambridge, United Kingdom; [2]Brain Information Communication Research Laboratory Group, Advanced Telecommunications Research Institute International, Kyoto, Japan; [3]Center for Information and Neural Networks, National Institute for Information and Communications Technology, Osaka, Japan; [4]Division of Anaesthesia, University of Cambridge, Cambridge, United Kingdom; [5]Behavioural and Clinical Neuroscience Institute, Department of Psychology, University of Cambridge, Cambridge, United Kingdom

**Abstract** Tonic pain after injury characterises a behavioural state that prioritises recovery. Although generally suppressing cognition and attention, tonic pain needs to allow effective relief learning to reduce the cause of the pain. Here, we describe a central learning circuit that supports learning of relief and concurrently suppresses the level of ongoing pain. We used computational modelling of behavioural, physiological and neuroimaging data in two experiments in which subjects learned to terminate tonic pain in static and dynamic escape-learning paradigms. In both studies, we show that active relief-seeking involves a reinforcement learning process manifest by error signals observed in the dorsal putamen. Critically, this system uses an uncertainty ('associability') signal detected in pregenual anterior cingulate cortex that both controls the relief learning rate, and endogenously and parametrically modulates the level of tonic pain. The results define a self-organising learning circuit that reduces ongoing pain when learning about potential relief.

DOI: https://doi.org/10.7554/eLife.31949.001

*For correspondence:
sz321@cam.ac.uk (SZ);
bjs49@cam.ac.uk (BS)

**Competing interests:** The authors declare that no competing interests exist.

## Introduction

Tonic pain is a common physiological consequence of injury and results in a behavioural state that favours quiescence and inactivity, prioritising energy conservation and optimising recuperation and tissue healing. This effect extends to cognition, and decreased attention is seen in a range of cognitive tasks during tonic pain (*Crombez et al., 1997*; *Lorenz and Bromm, 1997*). However, in some circumstances, this could be counter-productive, for instance if attentional resources were required for learning some means of relief or escape from the underlying cause of the pain. A natural solution would be to suppress tonic pain when relief learning is possible. Whether and how this is achieved is not known, but it is important as it might reveal central mechanisms of endogenous analgesia.

Two observations provide potential clues as to how a relief learning system might modulate pain. First, in some situations, perceived controllability has been found to reduce pain (*Salomons et al., 2004*; *Salomons et al., 2007*; *Wiech et al., 2014*; *Becker et al., 2015*), suggesting that the capacity to seek relief can engage endogenous modulation. Second, instructed attention has commonly been observed to reduce pain (*Bantick et al., 2002*). Therefore, it may be that attentional processes that are *internally* triggered when relief is learnable might provide a key signal that controls reduction of pain.

**eLife digest** Chronic pain lasting longer than three months is a common problem that affects about 1 in 5 people at some point in their lives. The lack of effective treatments has led to widespread use of a group of drugs called opioids – the best-known example is morphine. Opioids work by activating the brain's natural painkilling system and are useful to relieve short-term pain, for example in trauma or surgery, or in end-of-life care. Unfortunately, long-term use of opioids can cause many undesirable effects, including drug dependency. Misuse of opioids combined with the widespread availability of prescription drugs have contributed to the current crisis of opioid addiction and overdose.

A better understanding of how the brain's natural painkilling system works could help scientists develop painkillers that offer relief without the harmful side effects of opioids. While unpleasant, pain is important for survival. After an injury, for example, pain saps motivation and forces people to rest and preserve their energy as they are healing. In a way, this sort of pain is healthy because it promotes recovery. There may be times when the brain might want to turn off pain, such as when an individual is seeking new ways to relieve or manage pain. For example, by finding a way to cool a burn.

Now, Zhang et al. show that the brain reduces pain while individuals are trying to find relief. In the experiments, a metal probe was attached to the arm of healthy volunteers and heated until it became painful but not hot enough to burn the skin. Then, the volunteers were asked to play a game in which they had to find out which button on a small keypad cooled down the probe. Sometimes it was easy to turn off the heat, sometimes it was difficult. During the game, volunteers reported how much pain they felt and Zhang et al. used brain imaging to see what happened in their brains.

When the subjects were actively trying to work out which button they should press, pain was reduced. But when the subjects knew which button to press, it was not. Next, Zhang et al. found that a part of the brain called the pregenual cingulate cortex was responsible for making decisions about when to turn off pain and may so trigger the brain's natural pain killing system. A next step will be to see how this part of the brain decides to turn off pain and if it also controls opioid-like or other chemicals. This could improve the use of opioids, or even help to discover alternative treatments for chronic pain.

DOI: https://doi.org/10.7554/eLife.31949.002

In general, learning involves distinct processes of prediction ('state learning') and control ('action learning') (*Mackintosh, 1983*), although relief learning during tonic pain has not been thoroughly investigated. But a quantitative model of relief learning - one that describes the computational processes that are implemented in learning centres in the brain - would allow interrogation of how an attentional process might operate to modulate tonic pain. In the case of phasic pain, learning can be described by reinforcement learning (RL) models - a well-studied computational framework for learning from experience. RL models describe how to predict the occurrence of inherently salient events, and learn actions to exert control over them (maximising rewards, minimising penalties) (*Seymour et al., 2004*). RL models aim to provide a *mechanistic* (beyond a merely descriptive) account of the information processing operations that the brain actually implements (*Dayan and Abbott, 2001*), and have a solid foundation in classical theories of animal learning (*Mackintosh, 1983*). In such models, an agent learns state or action value functions through outcomes provided by interacting with the world. These functions can be learned by computing the error between predicted and actual outcomes, and using the error to improve future predictions and actions (*Sutton and Barto, 1998*). Experimentally, the validity of these models can be tested by comparing how well different model-generated predictors fit the actual behavioural and/or neural data (*O'Doherty et al., 2007*).

During learning, attention is thought to boost learning of predictive associations and suppress other irrelevant information. Computationally, this can be achieved by estimating the uncertainty as predictive associations are learned, and using this as a metric to control learning rates. Accordingly, high uncertainty corresponds to high attention and leads to more rapid learning (*Dayan et al.,*

*2000*; *Yu and Dayan, 2005*). One well-recognised way of formalising uncertainty in RL is by computing a quantity called the *associability*, which calculates the running average of the magnitude of recent prediction errors (i.e. frequent large prediction errors implies high uncertainty/associability). The concept of associability is grounded in classical theories of Pavlovian conditioning (the 'Pearce-Hall' learning rule, *Le Pelley, 2004*; *Pearce and Hall, 1980*; *Holland and Schiffino, 2016*), and provides a good account of behaviour and neural responses during Pavlovian learning (*Li et al., 2011*; *Boll et al., 2013*; *Zhang et al., 2016*). In this way, associability reflects a *computational* construct that captures aspects of the *psychological* construct of attention.

If it is the case, therefore, that attention can be understood as an uncertainty signal that drives learning during relief-seeking, it can then be tested with it modulates tonic pain in parallel. Standard models of RL do not include any mechanism by which the subjective experience of outcomes is under control, although in principle endogenous modulation of tonic pain could arise from any component of the learning system, including an associability signal. Using an associability signal in this way would make intuitive sense, because it would reduce ongoing pain when requirement for learning was high.

The studies presented here set two goals: to delineate the basic neural architecture of relief learning from tonic pain (i.e. pain escape learning) based on a state and action learning RL framework; and to understand the relationship between relief learning and endogenous pain modulation that is, to test the hypothesis that an attentional learning signal reduces pain. We studied behavioural, physiological and neural responses during two relief learning tasks in humans, involving (i) static and (ii) dynamic cue-relief contingencies. These tasks were designed to place a high precedence on error-based learning and uncertainty, as a robust test for learning mechanisms and dynamic modulation of tonic pain. Using a computationally motivated analysis approach, we aimed to identify whether behavioural and brain responses were well described as state and/or action RL learning systems and examined whether and how they exerted control over the perceived intensity of ongoing pain.

## Results

### Experiment 1

Experiment 1 was an escape learning task (n = 19) with fixed, probabilistic cue-relief contingencies (*Figure 1a*). Each subject performed three instrumental sessions and three Pavlovian sessions, to allow us to compare active and passive relief learning (*Figure 1b*). During each session (lasting approximatey 5 min), subjects were held in continuous pain by a thermal stimulator attached to their left arm, and temporary relief (i.e. escape) was given by rapidly cooling the thermode for 4 s, after which it returned to the baseline tonic pain level (*Figure 1c*). In instrumental sessions, subjects actively learned to select actions, a left or right button press, after viewing one of two visual cues (fractal images on a computer screen). For one of the cues, the probability of relief was 80% for one action and 20% for the other action, and for the other cue, the action relief probabilities were 60% and 40%. In the Pavlovian sessions, stimulus and outcome sequences were yoked to instrumental sessions for individual subjects, and subjects were required simply to press a button to match a random direction appearing on screen 0.5 s after visual cue onset (to control for motor responses). Subjective ratings of pain and relief were collected in random trials after outcome delivery, with on average eight pain and eight relief ratings per paradigm that is total 16 for each subject. All behavioural data including raw SCRs, choices, and ratings can be found in the manuscript data attachment.

#### Behavioural results
#### Choice

In instrumental learning, participants can learn which actions maximise the chance of relief. We assessed the ability of RL models to explain subjects' choice data, in comparison to a simple win-stay-lose-shift (WSLS) decision-making rule. We compared two basic RL models that have been widely studied in neurobiological investigations of reward and avoidance - a temporal difference (TD) action learning model with a fixed learning rate, and a version of the TD model with an adaptive learning rate based on action associabilities (hybrid TD model). As mentioned above, the

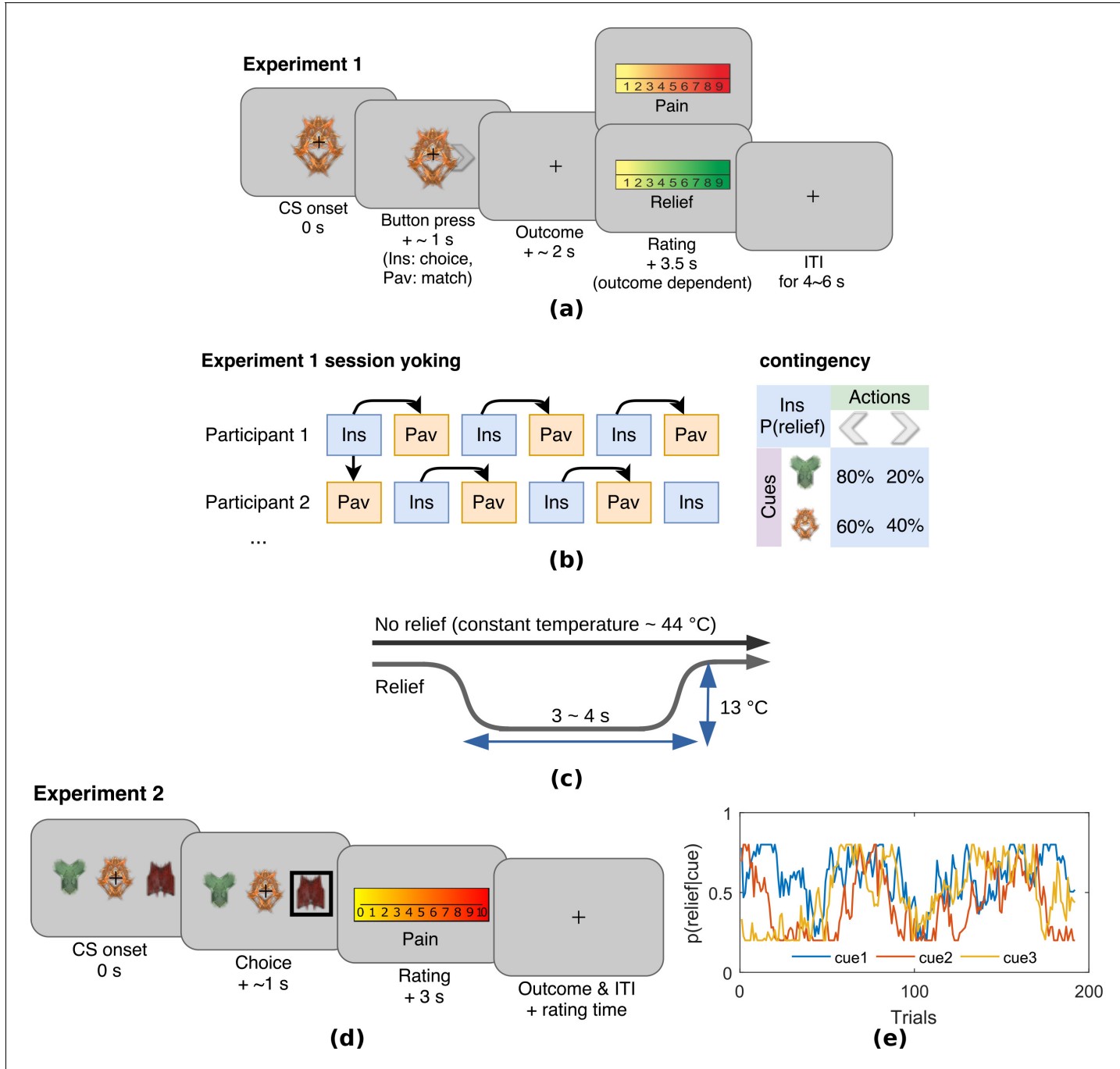

**Figure 1.** Experimental paradigms. (a) Example trial in Experiment 1, which was an instrumental relief learning task (Ins) with fixed relief probabilities, yoked with identical Pavlovian task (Pav) within subject. In instrumental trials, subjects saw one of two images ('cues') and then chose a left or right button press, with each action associated with a particular probability of relief. In the yoked Pavlovian session, subjects were simply asked to press button to match the action shown on screen (appearing 0.5 s after CS onset). (b) Instrumental/Pavlovian session yoking and cue-outcome contingency in Experiment 1, arrows represent identical stimulus-outcome sequence. Note in contingency table, left and right button presses were randomised for both actions and cues. (c) Relief and no relief outcomes, individually calibrated, constant temperatures at around 44°C were used to elicit tonic pain; a brief drop in temperature of 13°C was used as a relief outcome (4 s in Experiment 1, 3 s in Experiment 2), but temperature did not change for the duration in no relief outcomes. (d) Example trial in Experiment 2, where subjects performed an instrumental paradigm (only) involving unstable relief probabilities. The cue-action representation was different to Experiment 1, and three cues were presented alongside each other with subjects required to choose one of the three using a button press. The position of each cue varied from trial-to-trial, and the same three cues were presented throughout. Tonic pain rating being taken before the outcome was experienced, not after as in Experiment 1. (e) Example traces of dynamic relief

*Figure 1 continued on next page*

*Figure 1 continued*
probabilities for the three displayed cues throughout all trials in eight sessions in Experiment 2, which required a constant trade-off of exploration and exploitation throughout the task. Dynamic relief probabilities also provide varying uncertainty throughout learning.
DOI: https://doi.org/10.7554/eLife.31949.003

associability reflects the uncertainty in the action value, where higher associability indicates high uncertainty during learning, and is calculated based the recent average of the prediction error magnitude for each action. In a random-effects model comparison procedure (*Daunizeau et al., 2014*), we found that choices were best fit by the basic TD model (model frequency = 0.964, exceedance probability = 1, *Figure 2a*). Thus, there is no evidence that associability operates directly at the level of *actions*.

## Skin conductance responses (SCR)

To investigate physiological indices of learning, we examined trial-by-trial skin conductance responses (SCRs) during the 3 s cue time, before outcome presentation. SCRs obtained in instrumental sessions were higher compared to yoked Pavlovian sessions (*Figure 2b*, n = 15, see Materials and methods for session exclusion criteria, paired t-test T(14)=2.55, p=0.023), with the average SCR positively correlated between paradigms across individuals (Pearson correlation $\rho$=0.623, p=0.013, n = 15). Raw traces and cue-evoked responses of SCRs can be found in Figure supplements.

In Pavlovian aversive (fear) learning, SCRs have been shown to reflect the associability of Pavlovian predictions (*Li et al., 2011*; *Boll et al., 2013*; *Zhang et al., 2016*). Here, associability is calculated as the mean prediction error magnitude for the *state* (i.e. regardless of actions) (*Le Pelley, 2004*). In instrumental learning, Pavlovian learning of state-outcome contingencies still proceeds alongside action-outcome learning, distinct from instrumental choices, so Pavlovian state-outcome learning can be modelled in both instrumental and Pavlovian sessions. Consistent with previous studies of *phasic* pain, model-fitting revealed that a learning model with a state-based associability ('hybrid' model) best fit the SCR data in both Pavlovian and instrumental sessions (*Figure 2c* and *Figure 2d*, instrumental sessions: model frequency = 0.436, exceedance probability = 0.648, Pavlovian sessions: model frequency = 0.545, exceedance probability = 0.676), when tested against a competing simple Pavlovian Rescorla-Wagner model (akin to a TD model with only one state and a fixed learning rate). However, using the more stringent Protected Exceedance Probability analyses, the advantage of associability over other models were less conclusive (*Figure 2—figure supplement 3*). Together with the choice results, these analyses suggest that subjects use an associability-based RL mechanism for learning state values during both Pavlovian and instrumental pain escape, and a non-associability-based RL mechanism for learning action values in instrumental sessions. This divergence in learning strategies indicates that parallel learning systems coexist, which differ in their way of incorporating information about uncertainty in learning, as well as the nature of their behavioural responses.

## Ratings

Subjective ratings of pain and relief were taken intermittently after outcomes during the task, to explore how pain modulation might depend on relief learning. Ratings were taken on a sample of trials, so as to minimise disruption of task performance. Based on the fact that both controllability and attention are implicated in endogenous control, we hypothesised that pain would be reduced when the state-outcome associability was high, reflecting an attentional signal associated with enhanced learning. However, other types of modulation are possible. For instance, pain might be non-specifically reduced in instrumental, versus Pavlovian learning, reflecting a general effect of instrumental controllability. Alternatively, pain might be reduced by the expectation of relief that arises during learning, as it is known that conditioning alone can support placebo analgesia responses (*Colloca et al., 2008*) (although the extent to which this occurs might depend on the acquisition of contingency awareness during learning) (*Montgomery and Kirsch, 1997*; *Locher et al., 2017*). In this case, pain would be positively correlated with the relief prediction error, since it reports the difference between expectation and outcome.

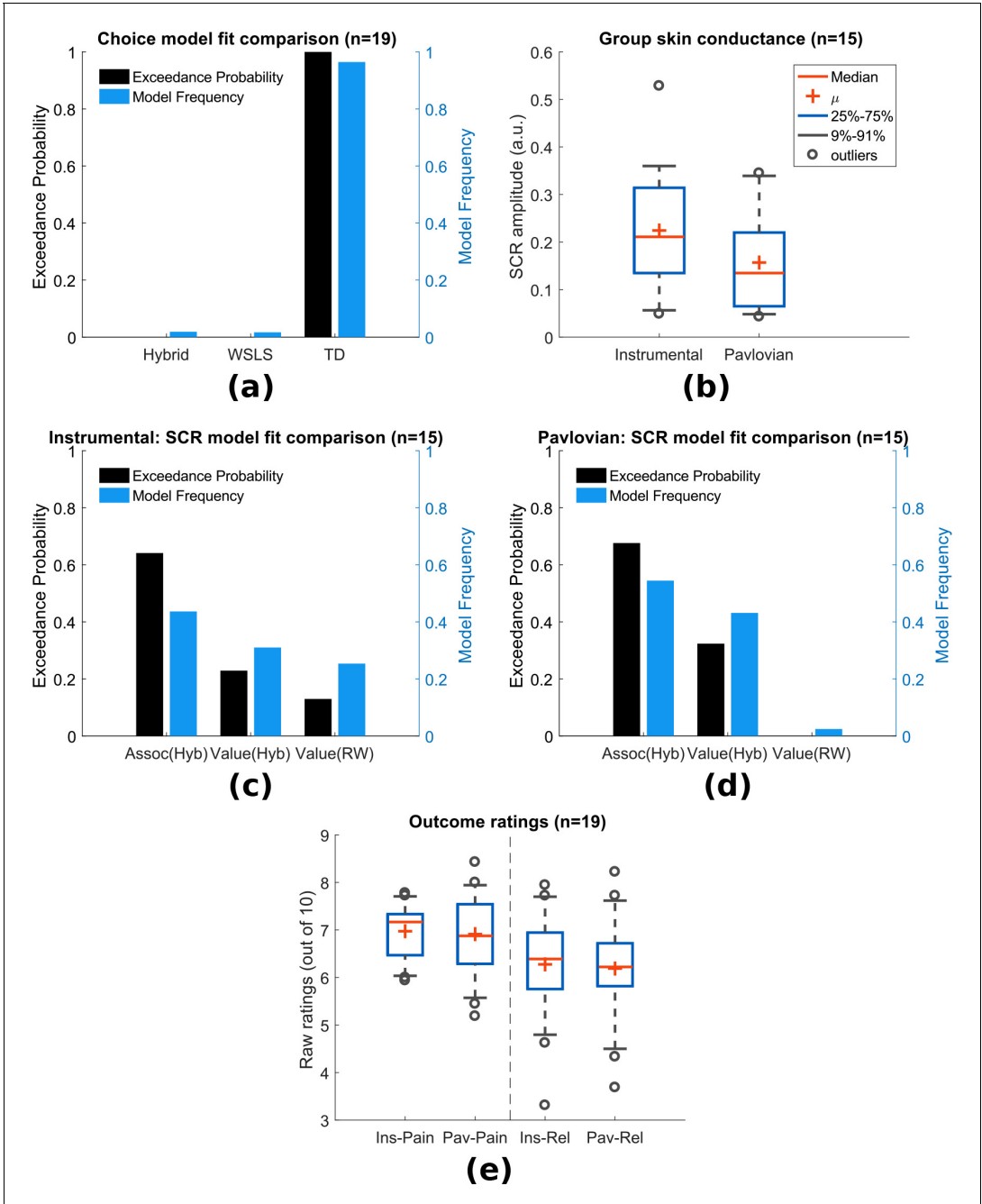

**Figure 2.** Experiment 1: behavioural results. (**a**) Choice-fitted model comparison, TD model fit instrumental sessions choices best (TD: action-learning model with fixed learning rate, Hybrid: action-learning model with associability as changing learning rate, WSLS: win-stay-lose-shift model). Model frequency represents how likely a model generate the data given a random participant, while exceedance probability estimates how one model is more likely compared to others (**Stephan et al., 2009**). (**b**) Instrumental vs Pavlovian sessions SCRs (n = 15, sessions with over 20% trials <0.02 amplitude excluded). (**c**) Associability from hybrid model fitted trial-by-trial SCRs best in instrumental sessions (Assoc: associability, Hyb: hybrid model, RW: Rescorla-Wagner model). (**d**) Associability also fitted SCRs from Pavlovian sessions best. (**e**) Both pain and relief ratings did not differ significantly between instrumental and Pavlovian sessions (Participants' ratings were averaged for each of the four categories shown, mean = 8 ratings per person per category).

DOI: https://doi.org/10.7554/eLife.31949.004

The following source data and figure supplements are available for figure 2:

**Source data 1.** Experiment 1's behavioural data including SCRs, choices, ratings can be found in zip file attached.

DOI: https://doi.org/10.7554/eLife.31949.008

*Figure 2 continued*

**Figure supplement 1.** Experiment 1: raw skin conductance traces, where vertical lines are beginning of each trial when cue display starts (n = 15, excluded participants not shown, showing first non-excluded session from all participant).

DOI: https://doi.org/10.7554/eLife.31949.005

**Figure supplement 2.** Experiment 1: filtered skin conductance traces (band-pass at 0.0159–2 Hz, 1 st order Butterworth), averaged across all trials within participant (n = 15, excluded participants not shown, shaded region represent SEM across all participants).

DOI: https://doi.org/10.7554/eLife.31949.006

**Figure supplement 3.** Experiment 1: Model protected exceedance probability.

DOI: https://doi.org/10.7554/eLife.31949.007

To test these competing hypotheses, we first compared the mean ratings of both pain (following a 'no relief' outcome) and relief (following a relief outcome) between Pavlovian and instrumental sessions, and found no significant difference (Mean±SEM, n = 19, mean = 8 ratings per person per category, instrumental pain: 6.97±0.13, Pavlovian pain: 6.91±0.20, instrumental relief: 6.46±0.24, Pavlovian relief: 6.33±0.27, between paradigm paired t-test both ratings p>0.5, *Figure 2e*). Hence, there is no support for a general effect of instrumental controllability on subjective pain and/or relief experience. We noted that mean pain and relief ratings were correlated with each other across individuals (ratings averaged across paradigms, Spearman's correlation $\rho$=0.73, p<0.001), indicating that higher perceived tonic heat pain was associated with higher cooling-related relief.

Next, we correlated pain ratings with the state-based associability and TD prediction error. In accordance with our hypothesis, in instrumental sessions associability was found to be negatively correlated with pain ratings (mean Spearman's $\bar{\rho}$=−0.177, one-sample t-test of Fisher's z-transformed correlation coefficients T(18)=-2.125, p=0.048). In Pavlovian sessions, however, we did not find a correlation ($\bar{\rho}$=−0.114, T(18)=0.758, p=0.458). There was no significant interaction between associability and paradigm (repeated measure ANOVA F(1,18)=1.247, p=0.279). This suggests that although associability is associated with pain modulation, this effect is not necessarily *specific* to instrumental sessions.

We found that the prediction errors were negatively correlated with pain ratings in Pavlovian sessions ($\bar{\rho}$=−0.356, T(18)=-3.198, p=0.005), but not instrumental sessions ($\bar{\rho}$=−0.154, T(18)=0.720, p=0.481). That is, when relief was omitted (i.e. as was always the case on the pain rating trial), a larger frustrated (i.e. negative) relief prediction error was associated with an increase in pain - in contrast to the prediction of a placebo expectation hypothesis. Finally, we also looked at relief ratings, but failed to find any significant correlation with either associability or prediction error in either instrumental or Pavlovian sessions.

## Neuroimaging results

The behavioural findings support the hypothesis that an associability signal that arises during state-based learning is associated with reduction of pain. Next, therefore, we then sought to identify (i) neural evidence for an error-based relief learning process and (ii) the neural correlates of the associability signal associated with tonic pain modulation. We implemented the TD action-learning model and associability-based hybrid TD state-learning model as determined from the behavioural data, using group-mean parameters (learning rate in TD model, and free parameter $\kappa$ and $\eta$ in hybrid TD model) to re-estimate trial-by-trial prediction errors/associability values for each subject as parametric modulators of fMRI BOLD time-series in general linear models.

## Prediction errors

The prediction error represents the core 'teaching' signal of the reinforcement learning model, and we specified *a priori* regions of interest based on the areas known to correlate with the prediction error in previous reinforcement learning studies of pain and reward (ventral and dorsal striatum, ventromedial prefrontal cortex (VMPFC), dorsolateral prefrontal cortex (DLPFC), and amygdala (*Seymour et al., 2005*; *Garrison et al., 2013*; *FitzGerald et al., 2012*)).

First, we looked for brain responses correlated with the action prediction error from the TD model in instrumental sessions. This identified responses in bilateral putamen, bilateral amygdala, left DLPFC, and VMPFC (*Figure 3a*, *Table 1*).

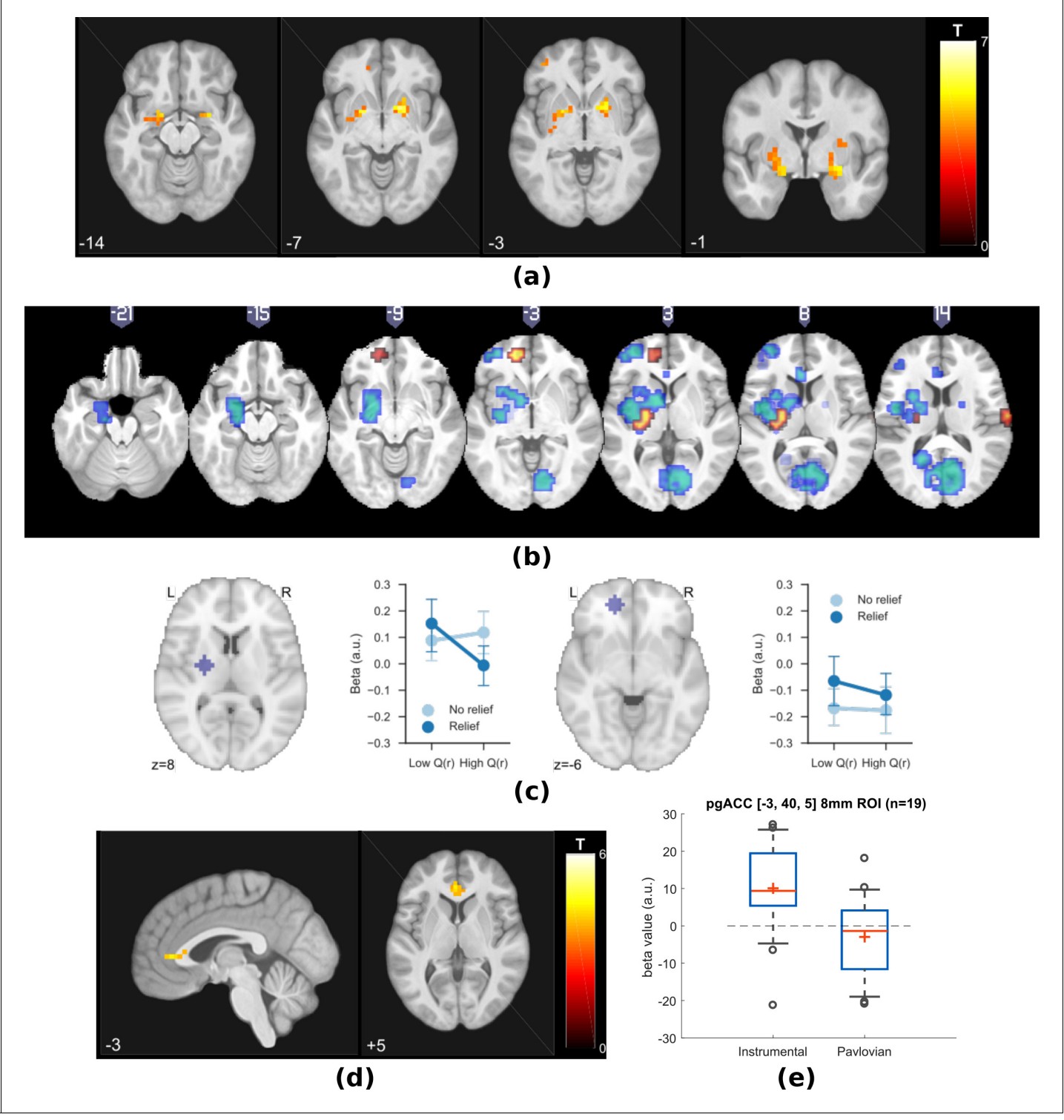

**Figure 3.** Experiment 1: neuroimaging results, shown at p<0.001 uncorrected: (**a**) TD model prediction errors (PE) as parametric modulators at outcome onset time (duration = 3 s). (**b**) Model PE posterior probability maps (PPMs) from group-level Bayesian model selection (BMS) within PE cluster mask, warm colour: TD model PE, cool colour: hybrid model PE (shown at exceedance probability P>0.7). (**c**) Axiomatic analysis of hybrid model PEs in instrumental sessions, ROIs were 8 mm spheres from BMS peaks favouring TD model PEs, in left putamen and VMPFC. (**d**) Associability uncertainty generated by hybrid model, as parametric modulators at choice time (duration = 0), in instrumental sessions. (**e**) Comparing pgACC activations across instrumental/Pavlovian paradigms, ROI was 8 mm sphere at [−3, 40, 5], peak from overlaying the pgACC clusters from Experiments 1 and 2.
DOI: https://doi.org/10.7554/eLife.31949.009

**Table 1.** Multiple correction for Experiment 1 (cluster-forming threshold of p<0.001 uncorrected, regions from Harvard-Oxford atlas. *FWE cluster-level corrected (showing p<0.05 only).

| p* | k | T | Z | MNI coordinates (mm) | | | Region mask |
|----|---|---|---|---|---|---|---|
| | | | | x | y | z | |
| TD model PE, instrumental sessions | | | | | | | |
| 0.007 | 4 | 4.27 | 3.5 | −21 | -5 | −14 | Amygdala L |
| 0.011 | 3 | 4.98 | 3.9 | 28 | -1 | −14 | Amygdala R |
| 0 | 28 | 5.31 | 4.07 | −21 | 3 | -7 | Putamen L |
| | | 4.7 | 3.75 | −28 | -5 | 1 | |
| 0.003 | 14 | 5.73 | 4.27 | 20 | 7 | -7 | Putamen R |
| 0.034 | 2 | 3.75 | 3.18 | 28 | -1 | 8 | |
| 0.007 | 4 | 4.63 | 3.71 | −17 | 3 | -3 | Pallidum L |
| 0.003 | 9 | 5.2 | 4.01 | 17 | 7 | -3 | Pallidum R |
| Hybrid model PE, instrumental sessions | | | | | | | |
| 0.005 | 5 | 4.3 | 3.52 | −21 | -5 | −14 | Amygdala L |
| 0.014 | 2 | 4.53 | 3.65 | 28 | -1 | −14 | Amygdala R |
| 0.004 | 12 | 5.02 | 3.92 | −21 | 3 | -7 | Putamen L |
| 0.012 | 6 | 4.55 | 3.66 | −28 | 3 | 8 | |
| 0.046 | 1 | 3.82 | 3.23 | −28 | 11 | -3 | |
| 0.001 | 23 | 5.03 | 3.92 | 20 | 7 | -7 | Putamen R |
| | | 4.92 | 3.87 | 20 | 7 | 1 | |
| | | 4.39 | 3.57 | 24 | -1 | 5 | |
| 0.006 | 5 | 4.04 | 3.36 | −17 | 3 | -3 | Pallidum L |
| 0.005 | 6 | 4.82 | 3.81 | 17 | 7 | 1 | Pallidum R |
| Hybrid model PE, Pavlovian sessions | | | | | | | |
| None | | | | | | | |
| Hybrid model associability, instrumental sessions | | | | | | | |
| 0.027 | 5 | 4.34 | 3.55 | -2 | 37 | 5 | Cingulate Anterior |

DOI: https://doi.org/10.7554/eLife.31949.010

Since action-outcome learning and state-outcome learning co-occur during instrumental sessions, we next modelled the state prediction error from the hybrid model in a separate regression model. In instrumental sessions, this revealed responses in similar regions to the TD action prediction error: in the striatum, right amygdala and left DLPFC (figure not shown, *Table 1*), consistent with the fact that state and action prediction errors are highly correlated.

To test which regions were better explained by each, we conducted a Bayesian model selection (BMS) within the prediction error ROIs (a conjunction mask of correlated clusters to both prediction error signals). This showed that the action-learning TD model had higher posterior and exceedance probabilities in the dorsal putamen, and VMPFC (*Figure 3b* warm colour clusters). The state-learning (hybrid) model better explained activities in the amygdala, ventral striatum, and DLPFC (*Figure 3b* cool colour clusters). Applying the same hybrid model prediction error signal in Pavlovian sessions only identified much weaker responses that did not survive multiple correction, in regions including the left amygdala (figure not shown) (*Table 1*).

To further illustrate the nature of the outcome response, we calculated a median split of the preceding cue values (based on the TD model), and looked at the outcome response for relief and no-relief outcomes. A prediction error response should be (i) higher for relief trials and (ii) higher when the preceding cue value was low (i.e. when relief was delivered when it was not expected) (*Roy et al., 2014*). As illustrated in *Figure 3c*, this 'axiomatic' analysis reveals some features of the prediction error, but lacks the resolution to illustrate it definitively.

## Associability

Since the behavioural data showed that the state-based associability correlated negatively with tonic pain ratings, we examined BOLD responses correlated with trial-by-trial associability from the hybrid model, by using the associability as a parametric regressor at the choice time (see Materials and methods for details of GLMs). We specified *a priori* ROIs according to regions previously implicated in attention and controllability-related endogenous analgesia, notably pregenual anterior cingulate cortex (pgACC), posterior insula and ventrolateral prefrontal cortex (VLPFC) (*Salomons et al., 2007*; *Wiech et al., 2006*); and associability (amygdala) (*Li et al., 2011*; *Zhang et al., 2016*; *Boll et al., 2013*).

We found correlated responses only in pgACC, in instrumental sessions (*Figure 3d*, *Table 1*, MNI coordinates of peak: [−2, 37, 5]). No significant responses were observed in Pavlovian sessions. *Figure 3e* illustrates individual subjects' beta values extracted from an 8 mm diameter spherical ROI mask built around peak coordinates [−3, 40, 5]. Instrumental sessions had higher response magnitude in pgACC compared to Pavlovian sessions across subjects (Instrumental sessions: one-sample t-test against 0 T(18)=3.746, p=0.0015, Pavlovian sessions: one-sample t-test against 0 T(18)=-1.230, p=0.235, paired t-test for instrumental versus Pavlovian T(18)=3.317, p=0.0038).

## Summary of experiment 1

In summary, the data indicate that (i) relief action learning is well described by a RL (TD) learning process, with action prediction error signals observed in the dorsal putamen, (ii) that state-outcome learning proceeds in parallel to action-outcome learning, and can be described by an associability-dependent hybrid TD learning mechanism, and (iii) that this state associability modulates the level of ongoing tonic pain during instrumental learning, with associated responses in pgACC.

This provides good evidence of a relief learning system that modulates pain according to learned uncertainty, and raises two important questions. First, can the associability signal be distinguished from other uncertainty signals that may arise in learning? Importantly, the use of fixed probabilities in the task means that associability tends to decline during sessions, raising the possibility that more complex models of uncertainty and attention might better explain the data, for instance those that involve changing beliefs that arise in changing (non-stationary) environments. Second, does the modulation of pain ratings occur throughout the trial? In the task, pain ratings are taken at the outcome of the action, and only when relief is frustrated, raising the possibility that it reflects an outcome-driven response, as opposed to learning-driven process modifying the ongoing pain. With these issues in mind, we designed a novel task to test if the model could be generalised to a different paradigm with greater demands on flexible learning.

# Experiment 2

In Experiment 2, 23 new subjects participated in a modified version of the instrumental escape learning task in Experiment 1, with a number of important differences. First, subjects performed only instrumental sessions (8 sessions with 24 trials in each) given the absence of a global effect of instrumental versus Pavlovian pain in the first experiment. Second, subjects were required to choose one out of *three* simultaneously presented visual cues to obtain relief, in which the position of each cue varied randomly from trial to trial. This was done to experimentally and theoretically better distinguish state-based and action-specific associability (*Figure 1d*). Third, the action-outcome contingencies were *non-stationary*, such that the relief probability from selecting each cue varied slowly throughout the experiment duration, controlled by a random walk algorithm which varied between 20 and 80% (*Figure 1e*). This ensured that associability varied constantly through the task, encouraging continued relief exploration, and allowed us to better resolve more complex models of uncertainty (see below). It also reduced the potential confounding correlation of associability and general habituation of SCRs. Fourth, we increased the frequency of tonic pain ratings (10 per session, 80 per subject in total) to enhance power for identifying modulatory effects on pain. Fifth, the rating was taken after the action but before outcome, to provide an improved assessment of ongoing tonic pain modulation without interference by the outcome. Finally, we also collected SCRs bilaterally, to enhance the data quality given the importance of the SCR in inferences about associability.

## Behavioural results

### Choice

In addition to the simple TD and hybrid action-learning TD models compared in Experiment 1, the modification in paradigm allowed us to test more sophisticated model-based learning models, including a hidden Markov model (HMM) (*Prévost et al., 2013*), and a hierarchical Bayesian model (*Mathys et al., 2011*). Both models incorporate a belief of environmental stability into learning, that is whether a cue previously predicting relief reliably has stopped being reliable during the course of the experiment. This is achieved by tracking the probability of state transition in the HMM, or environmental volatility in the hierarchical Bayesian model. Despite the greater demands of the non-stationary task compared to Experiment 1, the basic TD action learning model still best predicted choices following model comparison (model frequency = 0.624, exceedance probability = 0.989), followed by the HMM (model frequency = 0.192, exceedance probability = 0.006) and the hybrid action-learning model (model frequency = 0.174, exceedance probability = 0.004) (*Figure 4a*, see Methods for full details).

### SCR

SCRs were recorded from the side with thermal stimulation (left hand) and the side without stimulation (right hand). The left side had lower mean SCRs (*Figure 4b,L/R* paired t-test T(19)=-2.67, p=0.015, n = 20, exclusion criteria followed from Experiment 1), however, trial-by-trial SCRs were highly correlated between both sides within individual subjects (mean Pearson correlation $\bar{\rho}$=0.733, 18 out of 20 participants with p<0.001). This suggests that although the overall SCR amplitude might be suppressed by the tonic heat stimulus, this did not affect event-related responses.

Using the same model-fitting procedure as in Experiment 1 (with the addition that the model now predicted SCR on both hands for each trial), we found that the associability from the state-outcome hybrid model again provided the best fit of trial-by-trial SCRs (*Figure 4c*, model frequency = 0.667, exceedance probability = 0.954). Indeed, the associability-SCR fit has a much higher model exceedance probability compared with that in Experiment 1, presumably from including the less attenuated SCRs from the non-stimulated right side.

### Ratings

Experiment 1 suggested that the associability was correlated with modulation of tonic pain ratings. However, given the dynamic nature of Experiment 2, we investigated whether uncertainty measures related to other aspects of learning might offer a better account. To do this, we fitted multiple regression models to trial-by-trial ratings for each participant as follows:

$$\text{Rating} = \beta_1 \cdot \text{Relief} + \beta_2 \cdot \log(\text{Trial}) + \beta_3 \cdot \text{Predictor} \tag{1}$$

where the 'Relief' term is the number of trials since the previous relief outcome, $log(\text{Trial})$ is the log of trial number within session (1-24), 'Predictor' is the model generated uncertainty value. The 'Relief' and $log(\text{Trial})$ terms were included to account for potential temporal and sessional effects of the tonic pain stimulus.

We built a regression model with different uncertainty signals as predictors for comparison: the state-based associability from hybrid model (as in Experiment 1), the entropy of state-action posterior probabilities (approximate of uncertainty over values) in an HMM, the absolute value of prediction error from previous trial in TD model (as a model of surprise), and a null model that did not include 'Predictor' term (*Figure 4d*). In this analysis, the state-learning hybrid associability again best fit the pain ratings (model frequency = 0.698, exceedance probability = 0.980; n = 22, 1550 ratings, one participant was excluded for having >90% identical ratings). Regression coefficients with hybrid model associability as uncertainty predictor were significant across subjects (*Figure 4e*, one-sample t-test for three sets of coefficients: 'Relief' term: T(21)=-4.004, p<0.001 (i.e. habituation, reduced pain over time after relief), log(trial) term: T(21)=1.017, p=0.321, associability term: T(21)=-2.643, p=0.015).

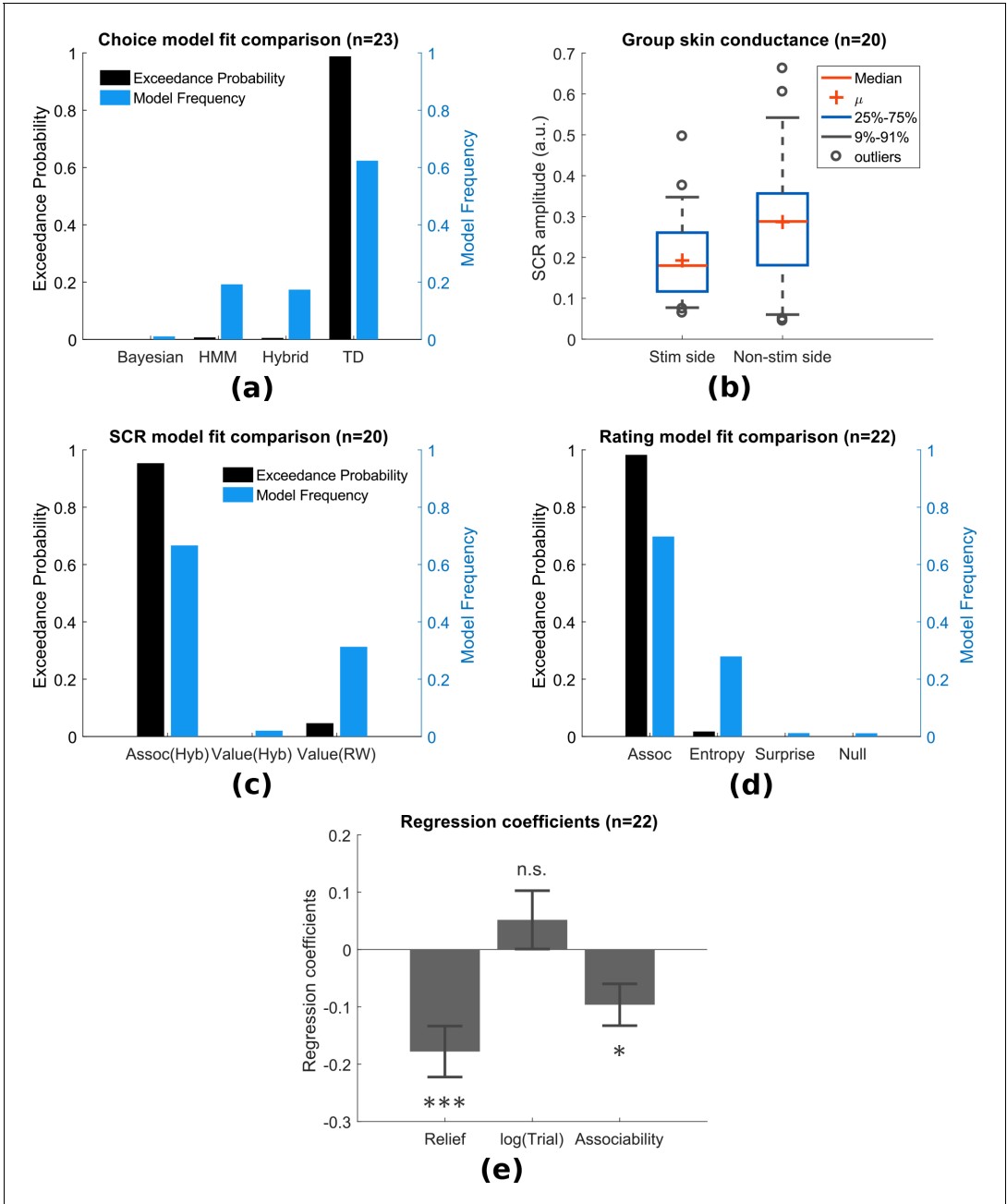

**Figure 4.** Experiment 2: behavioural results. (**a**) Model comparison showed that TD model fitted choices best (Bayesian: hierarchical Bayesian model, HMM: hidden Markov model, Hybrid: action-learning model with associability as changing learning rate). (**b**) SCRs measured on the side with thermal stimulation ('Stim side', left hand) were lower than those on without stimulation ('Non-stim side', right hand), but both were highly correlated. (**c**) Associability from state-learning hybrid model fit SCRs best, similarly to Experiment 1. (**d**) Trial-by-trial associability from hybrid model fitted pain ratings best compared with other uncertain measures (entropy: HMM entropy, surprise: TD model prediction error magnitude from previous trial, null model: regression with no predictors). (**e**) Regression coefficients with associability as uncertainty predictor were significantly negative across subjects.
DOI: https://doi.org/10.7554/eLife.31949.012

The following source data and figure supplements are available for figure 4:

**Source data 1.** Experiment 2: behavioural data including SCRs, choices, ratings can be found in zip file attached.
DOI: https://doi.org/10.7554/eLife.31949.016

**Figure supplement 1.** Experiment 2: raw skin conductance traces, where vertical lines are beginning of each trial when cue display starts (n = 20, excluded participants not shown, showing first non-excluded session from all participants).
DOI: https://doi.org/10.7554/eLife.31949.013

*Figure 4 continued on next page*

*Figure 4 continued*

**Figure supplement 2.** Experiment 2: filtered skin conductance traces (band-pass at 0.0159–2 Hz, 1 st order Butterworth), averaged across all trials within participant (n = 20, excluded participants not shown, shaded region represent SEM across all participants).
DOI: https://doi.org/10.7554/eLife.31949.014
**Figure supplement 3.** Experiment 2: model protected exceedance probability.
DOI: https://doi.org/10.7554/eLife.31949.015

## Neuroimaging results
### Prediction errors
We found that the TD model action prediction errors was robustly correlated with BOLD responses in similar regions identified in Experiment 1, including left dorsal putamen, bilateral amygdala, and left DLPFC (*Figure 5a*, *Table 2*). Of these, BMS showed the TD model had higher posterior and exceedance probabilities in the dorsal putamen, as well as amygdala and DLPFC (*Figure 5b* warm colour clusters). The state-learning hybrid model explained prediction error responses in several areas, but outside our original *a priori* regions of interest (see *Figure 5b* cool colour clusters).

As previously, we further illustrated the pattern of outcome responses as a function of preceding cue value and relief/no-relief in an 'axiomatic' analysis. We split the trial values into three bins, allowing a better inspection of responses permitted by our larger number of trials. This revealed a clear prediction error-like pattern in the dorsal putamen, but somewhat less clear cut in the amygdala and DLPFC (*Figure 5c*). Therefore, across all analysis methods and the two experiments, the left dorsal putamen robustly exhibited a response profile consistent with an escape-based relief prediction error.

### Associability
Following the same analysis as in Experiment 1, we found again that pgACC BOLD responses correlated with trial-by-trial associability from the state-learning hybrid model (*Figure 5d–e*, *Table 2*). The peak from this analysis was almost identical to that in Experiment 1 (Overlayed clusters can be found in Figure supplements). In addition, we used trial-by-trial pain ratings as a parametric modulator, but did not find significant pgACC responses, which suggested that it was unlikely to be solely driven by pain perception itself. Taken together, this indicates that the pgACC associability response is robust across experimental designs.

### Summary of experiment 2
In summary, Experiment 2 reproduced the main results of Experiment 1 within a non-stationary relief environment. Firstly, dorsal putamen correlated with an action-relief prediction error from the RL model. And secondly, pgACC correlated with a state-based associability signal, that in turn was associated with reduced tonic pain. In particular, this modulation of pain was present after the cue was presented (and not just at the outcome as in Experiment 1) and was better explained by the associability signal when compared against alternative uncertainty measures.

## Discussion
Across both experiments, the results provide convergent support for two key findings. First, we show that relief seeking from the state of tonic pain is supported by a reinforcement learning process, in which optimal escape actions are acquired using prediction error signals, which are observed as BOLD signals in the dorsal putamen. Second, we show that during learning, the level of ongoing pain is reduced by the learned associability associated with state-based relief predictions. This signal thus reduces pain when there is a greater capacity to learn new information and is associated with BOLD responses in the pregenual anterior cingulate cortex. Together, these results identify a learning circuit that governs tonic pain escape learning whilst also suppressing pain according to the precise information available during learning. In doing so, it solves the problem of balancing tonic pain with the requirement to actively learn about behaviour that could lead to relief.

The findings highlight the dual function of a state-based relief associability signal during tonic pain escape. Associability has its theoretical underpinnings in classical theories of associative learning and attention (i.e. the Pearce-Hall theory, *Pearce and Hall, 1980*), and its mathematical

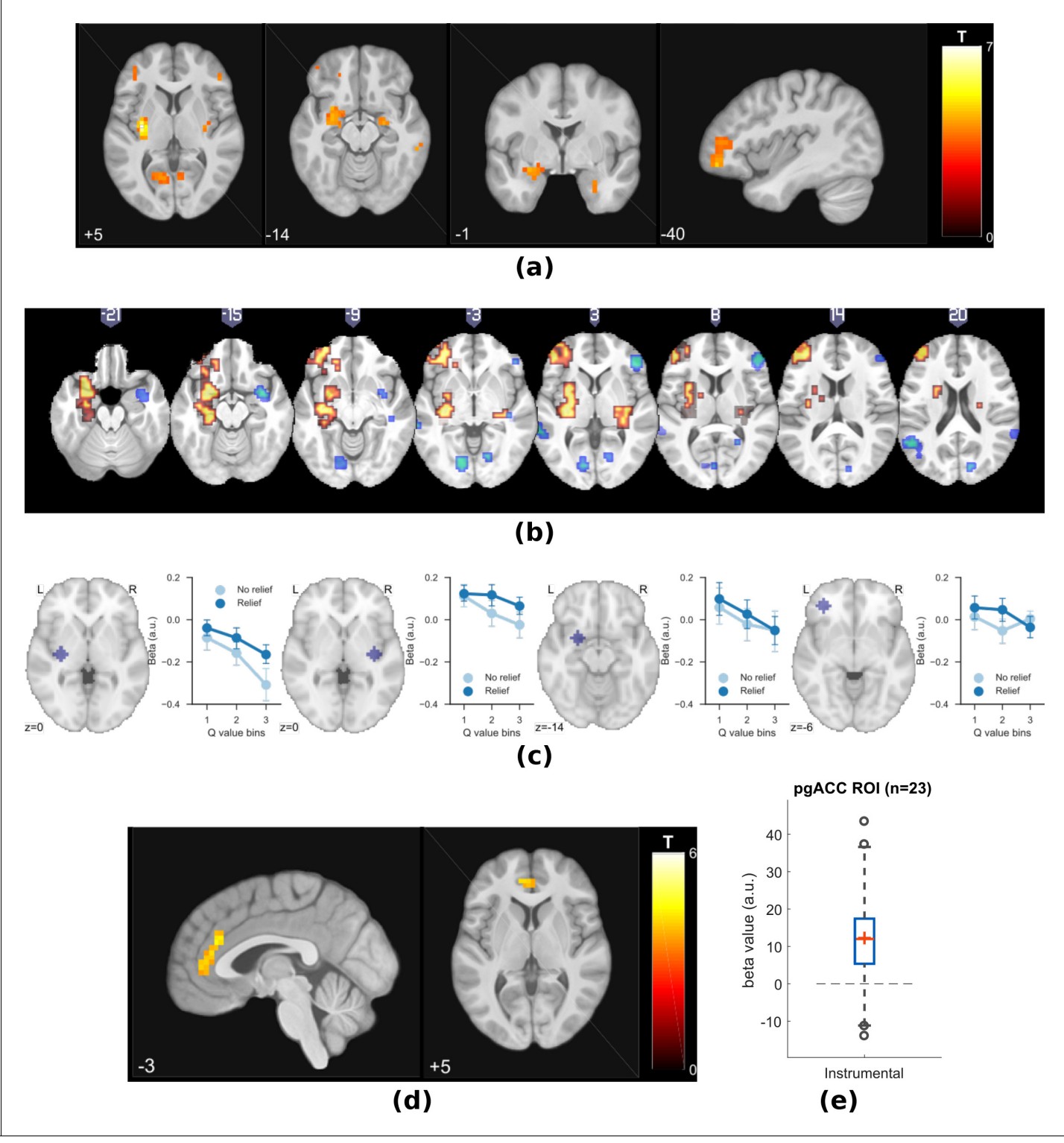

**Figure 5.** Experiment 2: neuroimaging results, shown at p<0.001 uncorrected: (a) TD model prediction errors (PE), at outcome onset time (duration = 3 s). (b) Model PE posterior probability maps (PPMs) from group-level Bayesian model selection, warm colour: TD model PE, cool colour: hybrid model PE (both shown at exceedance probability p>0.80). (c) Axiom analysis, separating trials according to outcomes and predicted relief values (bins 1–3 from low to high), BOLD activity pattern from striatum (putamen) satisfied those of relief PE. (d) Associability uncertainty generated by hybrid model correlating with pgACC activities, at choice time (duration = 0). (e) pgACC activation beta values across all subjects, ROI was 8 mm sphere at [−3, 40, 5], peak from overlaying the pgACC clusters from Experiments 1 and 2.

DOI: https://doi.org/10.7554/eLife.31949.018

*Figure 5 continued on next page*

*Figure 5 continued*

The following figure supplements are available for figure 5:

**Figure supplement 1.** Overlaying associability associated pgACC responses from both experiments (displayed at p<0.001 unc., crosshair at [−3, 40, 5]).
DOI: https://doi.org/10.7554/eLife.31949.019

**Figure supplement 2.** Overlaying prediction error associated responses from both experiments (displayed at p<0.001 unc., showing overlapping dorsal putamen and amygdala clusters).
DOI: https://doi.org/10.7554/eLife.31949.020

implementation here is as an approximate uncertainty quantity derived from computing the running average of the magnitude of the prediction error (*Sutton, 1992*; *Le Pelley, 2004*). This uncertainty signal effectively captures how predictable the environment is: when uncertainty is high (because of lots of recent large prediction errors), it increases the speed of acquisition through increasing the learning rate, and so accelerates convergence to stable predicted values. It is therefore an effective attention-like signal for mediating endogenous analgesia, because it selectively facilitates active relief seeking by suppressing pain only when it is necessary. This conception of the role of uncertainty in pain may explain why uncertainty has been shown to enhance phasic pain (*Yoshida et al., 2013*) - where pain acts as the signal to drive learning, and suppresses tonic pain, where pain acts to reduce general cognition. In both instances, the role of uncertainty and attention is to facilitate learning.

A caveat to this is that associability cannot distinguish *unreliable* cues - inherently poor predictors of outcomes, and so does not discriminate between reducible and irreducible uncertainty, bearing in mind there is little adaptive logic in suppressing pain for unreliable predictors. Over extended timeframes, it is possible that the learning system recognises this and reduces endogenous control. However, in rodent studies of associative learning, associability is maintained even after several days of training (*Holland et al., 2002*), and it is possible that salient cues in aversive situations maintain the ability to command attention and learning longer than that would be predicted by 'optimal' Bayesian models.

**Table 2.** Multiple correction for Experiment 2 (cluster-forming threshold of p<0.001 uncorrected, regions from Harvard-Oxford atlas. *FWE cluster-level corrected (showing p<0.05 only).

| p* | k | T | Z | MNI coordinates (mm) | | | Region mask |
|---|---|---|---|---|---|---|---|
| | | | | x | y | z | |
| TD model PE | | | | | | | |
| 0.002 | 15 | 4.31 | 3.63 | −25 | -5 | −22 | Amygdala L |
| 0.003 | 11 | 4.36 | 3.66 | 24 | -8 | −14 | Amygdala R |
| 0.018 | 1 | 3.97 | 3.41 | 28 | -1 | −26 | |
| 0.002 | 22 | 5.9 | 4.52 | −32 | -8 | 5 | Putamen L |
| 0.021 | 4 | 4.55 | 3.78 | 32 | −16 | 1 | Putamen R |
| Hybrid model PE | | | | | | | |
| 0.001 | 16 | 4.36 | 3.66 | −21 | −12 | −14 | Amygdala L |
| | | 4.23 | 3.58 | −21 | -1 | −18 | |
| 0.002 | 13 | 4.95 | 4.01 | 24 | -8 | −18 | Amygdala R |
| | | 4.34 | 3.65 | 28 | -1 | −26 | |
| 0.003 | 17 | 5.49 | 4.31 | −32 | -8 | 5 | Putamen L |
| Hybrid model associability | | | | | | | |
| 0.001 | 29 | 4.5 | 3.75 | -6 | 40 | 12 | Cingulate Anterior |
| | | 4.44 | 3.71 | -2 | 33 | 23 | |
| | | 4.08 | 3.49 | -2 | 44 | 5 | |
| | | 3.93 | 3.38 | 2 | 40 | 1 | |

DOI: https://doi.org/10.7554/eLife.31949.021

The localisation of the associability signal to the pgACC is consistent with *a priori* predictions. The region is known to be involved in threat unpredictability (*Rubio et al., 2015*; *Nitschke et al., 2006*), computations of uncertainty during difficult approach-avoidance decision-making (*Amemori and Graybiel, 2012*), and in the perseverance of behaviour during foraging (*McGuire and Kable, 2015*; *Kolling et al., 2012*). It is distinct from a more anterior region in the ventromedial prefrontal cortex associated with action value (*FitzGerald et al., 2012*). More importantly, it has been specifically implicated in various forms of endogenous analgesia, including coping with uncontrollable pain (*Salomons et al., 2007*), distraction (*Valet et al., 2004*), and placebo analgesia (*Bingel et al., 2006*; *Eippert et al., 2009*). However, an open question remains about the role of conscious awareness in driving pgACC-related endogenous control - a factor that is often important in these other paradigms. Whether or not the role of associability is modulated by the metacognitive awareness of uncertainty or controllability would be an important question for future studies.

The pgACC has been suggested to be central to a 'medial pain system' and the descending control of pain, with its known anatomical and functional connectivity to key regions including the amygdala (*Derbyshire et al., 1997*; *Vogt et al., 2005*; *Salomons et al., 2015*) and PAG (*Stein et al., 2012*; *Buchanan et al., 1994*; *Vogt, 2005*; *Domesick, 1969*). Evidence of high level of $\mu$-opioid receptors within pgACC (*Vogt et al., 2005*), where increased occupation has been found in both acute and chronic pain (*Zubieta et al., 2005*; *Jones et al., 2004*), further illustrates pgACC's potential role for cortical control of pain.

The results provide a formal computational framework that brings together theories of pain attention, controllability and endogenous analgesia. Previous demonstrations of reduced pain (albeit typically for phasic, not tonic pain) have been inconsistent (*Becker et al., 2015*; *Salomons et al., 2004*; *Salomons et al., 2007*; *Wiech et al., 2014*; *Wiech et al., 2006*; *Mohr et al., 2012*). Our results offer insight into why - by suggesting that endogenous analgesia is not a non-specific manifestation of control, but rather a specific process linked to the learnable information.

From the perspective of animal learning theory, the experiments here show how motivation during the persistent pain state can be understood as an escape learning problem, in which the state of relief is determined by the offset of a tonic aversive state (*Mackintosh, 1983*; *Solomon and Corbit, 1974*). This is theoretically distinct from the better-studied form of relief that results from *omission* of otherwise expected pain or punishment (*Konorski, 1967*), and which motivates avoidance behaviour (*Mowrer, 1960*). In our task, acquisition of dissociable behavioural responses (SCRs and choices) reveals the underlying theoretical architecture of the escape learning process, which involves both parallel state-outcome and action-outcome learning components. The action-outcome learning error signal localises to a region of the dorsolateral striatum (dorsal putamen). Striatal error signals are seen across a broad range of action learning tasks, although the region here appears more dorsolateral than previously noted in avoidance learning (*Kim et al., 2006*; *Seymour et al., 2012*; *Delgado et al., 2009*). It is not possible to definitively identify whether avoidance and escape use distinct errors, but it is well recognised that there are multiple error signals in dorsal and ventral striatum, for instance reflecting 'model-based' (cognitive), 'model-free' (including stimulus-response habits) and Pavlovian control (*Tricomi et al., 2009*; *Schonberg et al., 2010*; *Yin et al., 2004*). The reinforcement learning model we describe is a 'model-free' mechanism, since it learns action values but does not build an internal model of state-outcome identities and transition probabilities (*Daw et al., 2005*). However, it is likely that a model-based system co-exists and might be identifiable with appropriate task designs (*Daw et al., 2011*).

Developing a computational account of relief learning and endogenous control may also help us understand how the brain contributes to the pathogenesis and maintenance of chronic pain (*Navratilova and Porreca, 2014*). Adaptive learning processes are thought to be important in chronic pain: learning and controllability have been proposed to play a role in the pathogenesis and maintenance of chronic pain (*Vlaeyen, 2015*; *Flor et al., 2002*; *Apkarian et al., 2004*; *Salomons et al., 2015*), and brain regions such as the medial prefrontal cortex and striatum have been consistently implicated in clinical studies, for example in pain offset responses (*Baliki et al., 2010*) and resting functional connectivity in chronic back pain (*Baliki et al., 2008*; *Baliki et al., 2012*; *Fritz et al., 2016*; *Yu et al., 2014*). In addition to suggesting a possible computational mechanism that might underlie pain susceptibility in these patients, the results highlight the pgACC as a potential target for therapeutic intervention.

# Materials and methods

## Subjects
Two separate groups of healthy subjects participated in the two neuroimaging experiments (Experiment 1: n = 19, six female, age 26.1±5.1 years; Experiment 2: n = 23, five female, age 23.9±3.1 years). All subjects gave informed consent prior to participation, had normal or corrected to normal vision, and were free of pain conditions or pain medications. The two experiments were performed in different institutes, and approved by their relevant ethics and Safety committees: for the National Institute of Information and Communications Technology, Japan (Experiment 1), and the Advanced Telecommunications Research Institute, Japan (Experiment 2).

## Experimental design

### Experiment 1
Subjects participated in an interleaved instrumental conditioning and yoked Pavlovian relief conditioning sessions in which they actively or passively escaped from tonic pain, respectively. Tonic pain was maintained by constant thermal stimulation to the left inner forearm (see 'Stimulation' for details), and relief was induced by temporarily cooling the heat stimulus, which abolishes pain and causes a strong sense of relief.

In the instrumental conditioning sessions, subjects learned to select actions based on different cues. The cues were abstract fractal images on a computer screen. Actions were left or right button-presses on a response pad, and successful outcomes were the brief cooling (relief) period from the tonic painful heat. There were two types of visual cue: an 'easy' cue with high probability of relief when paired with a particular response (80% relief chance with one of the button press responses and 20% chance with the other response), and a 'hard' visual cue with a lower probability of relief with a particular response (60%/40% relief chance for the two response actions). These different outcome probabilities were used to induce experimental variability in the uncertainty of relief prediction. On each trial, the visual cue (conditioned stimulus, CS) appeared on screen for 3 s, during which subjects were asked to make the left or right button press response. An arrow corresponding to the chosen direction was superimposed on the cue after the decision was made until the 3 s display period ended. The disappearance of the cue and response arrow was followed immediately by the outcome of a temporary decrease in temperature of the painful heat stimulus (temporary reduction of temperature by 13°C from the tonic level for 4 s), or no change in temperature such that the constant pain continued straight on into the next trial. The next trial started after a jittered inter-trial interval (ITI) of 4–6 s (mean = 5 s) after outcome presentation concluded (*Figure 1a*). There were 20 trials per session, with equal number of 'easy' and 'hard' cues (n = 10 each). Each session lasted about 5 min.

The yoked Pavlovian conditioning task was identical to the instrumental task, except subjects did not have control over the outcomes through their responses. Instead, the sequence of cues and outcomes from the previous instrumental session were used (or the first instrumental session from the previous subject, for subjects who started with a Pavlovian session), although subjects were not aware of the yoking process. A different set of fractal images was used for the yoked Pavlovian sessions, so learning from an instrumental session could not be transferred to its corresponding Pavlovian session. To control for motor responses in both sessions, subjects were asked to press the response button according to the randomised indicator arrow, which appeared on screen 0.5 s after CS presentation. This is common in neuroimaging studies of Pavlovian and instrumental learning, and it was clearly explained to subjects that these actions bore no relationship to outcomes.

Each subject repeated instrumental and yoked Pavlovian sessions three times (six sessions in total). They were clearly instructed whether it was a Pavlovian or instrumental session. To remove any order confounds, the session order was alternated within and between subjects (i.e. order ABA-BAB, or BABABA), with half the subjects started with the instrumental and the other half with the Pavlovian task. A short break was taken every two sessions to allow the experimenter to change the location of the heat stimuli probe, to minimise effects of habituation/sensitization across the whole experiment.

Subjective ratings of perceived trial outcomes (pain relief or ongoing pain) were collected near the beginning, middle, and end of each session, in identical order for instrumental and its yoked

Pavlovian counterpart. A 0–10 rating scale appeared 3.5 s after outcome presentation (0.5 s overlap with relief duration if any), where the scale ranged from 0 (no pain at all) to 10 (unbearable pain) for no relief outcome (red scale in *Figure 1a*), and 0 (no relief at all) to 10 (very pleasant relief) for relief outcome (green scale). Although it is the case that ratings are inherently subjective, their modulation reflects an objective process that may explain a component of this apparent subjectivity. This does raise the issue of whether the subjective relief ratings influence the outcome values when learned in the RL model, but this (presumably subtle) effect is something that is beyond the experimental power of this experiments to resolve.

### Experiment 2

Experiment 2 was a purely instrumental relief conditioning task, similar to that of Experiment 1. However in this task, three visual cues were presented on screen simultaneously for 3 s, during which the subject was asked to choose one (*Figure 1d*) with a three-button response pad. Each one of these cues had varying relief probability, generated by a random walk process (probabilities changing at step size of 0.1, bound between 0.2 and 0.8, with random start). Relief outcomes were identical to that in Experiment 1, except the duration was reduced to 3 s, which was enough to produce a similar relief sensation with lower trial time. Subjects repeated the same task for eight sessions (24 trials each), with the same visual cues throughout. However, several subjects did not complete all sessions because of excess time in SCR experimental set-up which reduced the time available for the task; hence, the overall average was 7.08±1.44 sessions per subject.

Subjective pain ratings were collected after the 3 s choice period and before outcome presentation, in 10 random trials out of 24 in each session, with the same 0–10 rating scale in Experiment 1 (red scale only). We have summarised the details of ratings from both experiments in *Table 3*.

## Stimulation

Painful tonic thermal stimuli were delivered to the subject's skin surface above the wrist on the left inner forearm, through a contact heat-evoked potential stimulator (CHEPS, Medoc Pathway, Israel). The CHEPS thermode is capable of rapid cooling at 40°C/s, which made rapid temporary pain relief possible in an event-related design.

The temperature of painful tonic stimuli was set according to the subject's own pain threshold calibrated beforehand. In Experiment 1, before the task, two series of 6 pre-set temperatures were presented in random order (set 1: mean ± std 43.7±1.7°C; set 2: 44.6±0.6°C), with each temperature delivered for 8 s, after which the subject determined whether the stimulation period was painful or not (ISI = 8 s). The higher of the two lowest painful temperatures from the two tests was used as the tonic stimulation temperature.

In Experiment 2, 10 temperatures were presented in each series, both were randomly generated with 44.4±0.7°C. After the 8 s stimulation, subjects were asked to rate their pain on a 0–10 VAS scale, which were fitted with a sigmoid function. The temperature was chosen from the temperature range of: 44, 44.2, 44.5, 44.8, 45°C, whichever closest and below the model fitted value of VAS = 8.

The final temperature used did not differ hugely for the two experiments despite the change in thresholding method (Experiment 1: 44.3±0.2°C, Experiment 2: 44.5±0.4°C). The relief temperature was set constant at 13°C below threshold temperature for all subjects.

**Table 3.** Details of subjective ratings for Experiments 1 and 2.

| Experiment | Rating type | Rating timing | Avg # of ratings per subject |
|---|---|---|---|
| Experiment 1 | Instrumental pain | After 3 s cue + choice window AND outcome (rating type depend on outcome) | 8.2 |
| | Instrumental relief | | 7.7 |
| | Pavlovian pain | | 8.1 |
| | Pavlovian relief | | 7.7 |
| Experiment 2 | Instrumental pain | After 3 s cue + choice window, BEFORE outcome | 70.9 |

DOI: https://doi.org/10.7554/eLife.31949.022

## Physiological measures

Skin conductance responses (SCRs) were measured using MRI-compatible BrainAmp ExG MR System (Brain Products, Munich, Germany) with Ag/AgCl sintered MR electrodes, filled with skin conductance electrode paste.

In Experiment 1, SCR data were recorded on volar surfaces of distal phalanges of the second and fourth fingers on the left (tonic pain side with thermode attached). In Experiment 2, data were recorded from both hands, in the same location on the left (with thermode), and on the hypothenar eminences of the palm on the right (button press hand without thermode), with electrodes approximately 2 cm apart. The signals were collected using BrainVision software at 500 Hz with no filter.

Off-line processing and analysis were implemented in MATLAB7 (The MathWorks Inc., Natick, MA), with the PsPM toolbox (http://pspm.sourceforge.net/). Data were down-sampled to 10 Hz, band-pass filtered at 0.0159–2 Hz (1 st order Butterworth). Given the variable nature of SCR onset and duration in a learning experiment, the non-linear model in PsPM was used. Boxcar regressors were constructed at cue onset (duration = 3 s, cue presentation). These regressors were convolved with the canonical skin conductance response function, to estimate event-related response amplitude, latency, and dispersion (only SCR amplitude were used in modelling).

Sessions with more than 20% trials (4 out of 20 trials for Experiment 1, 5 out of 24 for Experiment 2) with cue-evoked SCR amplitude below the threshold of 0.02 were labelled as not having enough viable event related SCRs. In Experiment 1, 15 subjects and 50 sessions remained. In Experiment 2, 19 subjects and 79 sessions remained for the left (thermal stimulation side), 20 subjects and 96 sessions remained for the right (no stimulation side). For model fitting, right side SCR reject criteria were used, since both channel's data were included as two data sources. Trial SCRs were log-transformed within subject before model fitting. Transformed SCRs on both sides were highly correlated (*Figure 4b*).

## Other behavioural measures

Trial-by-trial choice data (button press indicating choices) and reaction times (length of time taken from CS onset to button press) of subjects were recorded as part of behavioural measurements.

## Computational learning models

To capture relief learning we fitted behavioural responses using different learning models from previous studies (*Table 4*). Free energy (F) are variational Bayesian approximation of model's marginal likelihood, table showing the sum of F for all participants to provide model absolute fit evaluation. Actual model comparison was conducted based on random-effect analysis. For instrumental learning, the reinforcement of subjects' responses (i.e. choices) based on relief experience can be modelled using reinforcement learning model (*Sutton and Barto, 1998*). For Pavlovian learning, physiological responses can be used for model fitting (*Li et al., 2011*; *Boll et al., 2013*; *Zhang et al., 2016*).

### Win-Stay-Lose-Shift (WLSL) model

WSLS assumes a subject has fixed pseudo Q values for each state-action pair, where a relief outcome always produces a positive value for the chosen state-action pair (i.e. win-stay), while the remaining state-action combinations had negative values (i.e. lose-shift). A no relief outcome flipped the sign of all values. Two free parameters $p_1$ and $p_2$ ($0 \leq p_{1,2} \leq 1$) scaling the pseudo Q values for the two cues presented were used in model fitting, which were assumed fixed throughout the experiment but varied for individuals.

### TD model

The predicted state-action value $Q$ given particular state $s$ and action $a$ between successive trials is updated using an error-driven delta rule with learning rate $\alpha$ ($0 \leq \alpha \leq 1$) (*Gläscher et al., 2010*; *Morris et al., 2006*; *Sutton and Barto, 1998*):

$$Q_{t+1}(s,a) = Q_t(s,a) + \alpha \cdot (r_t - Q_t(s,a)) \tag{2}$$

**Table 4.** All learning models fitted (bold: winning model; AL - action-learning; SL - state-learning, F - variational Bayesian approximation to the model's marginal likelihood, used for model comparison)

| Experiment 1 (Instrumental sessions) | | | |
| --- | --- | --- | --- |
| Choice | F (n=19, sum [sem]) | SCR | F (n = 15, sum [sem]) |
| **TD** | -1330.920 [3.604] | RW - value | −1079.153 [8.024] |
| Hybrid (AL) | -1345.667 [3.664] | Hybrid (SL) - value | −1077.911 [8.059] |
| WSLS | -1486.723 [3.973] | **Hybrid (SL) - associability** | −1077.699 [8.003] |
| Experiment 1 (Pavlovian sessions) | | | |
| Choice (not available) | | SCR | F (n = 15, sum [sem]) |
| N/A | | RW - value | −1101.079 [7.132] |
| | | Hybrid (SL) - value | −1096.250 [7.195] |
| | | **Hybrid (SL) - associability** | −1095.135 [7.106] |
| Experiment 2 (Instrumental sessions, Pavlovian not available) | | | |
| Choice | F (n=23, sum [sem]) | SCR | F (n = 20, sum [sem]) |
| **TD** | -3572.476 [8.736] | RW - value | −7867.834 [60.668] |
| Hybrid (AL) | -3626.478 [8.946] | Hybrid (SL) - value | −7857.341 [60.643] |
| HMM | -3571.020 [9.067] | **Hybrid (SL) - associability** | −7841.864 [60.838] |
| Bayesian Hierarchical | -3784.372 [8.616] | | |

DOI: https://doi.org/10.7554/eLife.31949.023

where $r_t$ is the outcome of the trial (relief = 1, no relief = 0). The probability of choosing action $a$ from a set of all available actions $A_s \in \{a, b, c...\}$ in trial $t$ is modelled by a softmax distribution,

$$p(a|s) = \frac{exp(\tau \cdot Q_t(s, a))}{\sum_{b \in A_s} exp(\tau \cdot Q_t(s, b))} \tag{3}$$

where $\tau$ is the inverse temperature parameter governing the competition between actions ($\tau > 0$).

## Rescorla-Wagner (RW) model

For Pavlovian learning, where choice decisions are not available, the standard temporal difference (TD) model updates the state value $V(s)$ based on prediction errors following the Rescorla-Wagner learning rule:

$$V_{t+1}(s) = V_t(s) + \alpha \cdot (r_t - V_t(s)) \tag{4}$$

## Hybrid model

The hybrid model incorporated an associability term as a changing learning rate for a standard TD model in value learning (*Le Pelley, 2004*; *Li et al., 2011*). The associability term is also referred to as Pearce-Hall associability, an equivalent measure of attention or uncertainty, which is modulated by the magnitude of recent prediction error. The varying learning rate can be used in Pavlovian state-learning:

$$V_{t+1}(s) = V_t(s) + \kappa \cdot \alpha_t(s) \cdot (r_t - V_t(s)) \tag{5}$$

$$\alpha_{t+1}(s) = \eta \cdot |r_t - V_t(s)| + (1 - \eta) \cdot \alpha_t(s) \tag{6}$$

where $\eta$, $\kappa$ are free parameters limited within the range of [0,1].

The model can also be extended to instrumental action-learning:

$$Q_{t+1}(s, a) = Q_t(s, a) + \kappa \cdot \alpha_t(s, a) \cdot (r_t - Q_t(s, a)) \tag{7}$$

$$\alpha_{t+1}(s,a) = \eta \cdot |r_t - Q_t(s,a)| + (1-\eta) \cdot \alpha_t(s,a) \tag{8}$$

## Hidden Markov Model (HMM)

For Experiment 2, where relief probability is unstable, model-based learning models were fitted to behavioural data. Hidden Markov Model with dynamic expectation of change (*Prévost et al., 2013*; *Schlagenhauf et al., 2014*) was adapted to incorporate a hidden state variable $S_t$ that represents the subject's estimation of an action-outcome pair (e.g. in Experiment 2, $S_t = (cue, relief)$, three cues × relief/no relief = 6 combinations). The state transition probabilities are calculated as:

$$P(S_t|S_{t-1}) = \begin{pmatrix} 1-\beta & \beta \\ \beta & 1-\beta \end{pmatrix} \tag{9}$$

where $\beta$ is a free parameter ($0 \le \beta \le 1$). For each cue, the symmetry of the transition matrix encodes the reciprocal relationship between relief/no relief belief. Given the hidden state variable, the probability of actually observing this outcome is updated as:

$$P(O_t|S_t) = 0.5 \times \begin{pmatrix} 1+c & 1-c \\ 1-d & 1+d \end{pmatrix} \tag{10}$$

where the rows of the matrix represent relief/no relief outcomes, the columns represent the relief/no relief belief in $S_t$. $c$ and $d$ are free parameters ($0 \le c \le 1$, $0 \le d \le 1$) to incorporate potential discrimination between the two outcome types. The prior probability of $S_t$ is calculated from the state transition probabilities and the posterior probability of $S_{t-1}$ (*Equation 11*). The posterior probability of $S_t$ is calculated from the prior $P(S_t)$ (from *Equation 11*) and the observed outcome $O_t$ (*Equation 12*):

$$P(S_t) = \sum_{S_{t-1}} P(S_t|S_{t-1})P(S_{t-1}) \tag{11}$$

$$P(S_t) = \frac{P(O_t|S_t)P(S_t)}{\sum_{S_t} P(O_t|S_t)P(S_t)} \tag{12}$$

where *Equation 11* is updated before observed outcome $O_t$, *Equation 12* is updated after $O_t$.

$S_t$ can be used to approximate state values by calculating the relative relief belief through a sigmoid function, with a free parameter $m$, and the preferred action to be inferred using the softmax function.

$$P(\mathrm{r}=1|\mathrm{cue}) = \frac{1}{1+\exp(-\mathrm{x})} \tag{13}$$

where $x = S_t(\mathrm{r}=1) - S_t(\mathrm{r}=0) + \mathrm{m}$.

To represent uncertainty under $i$ possible posterior relief probabilities, entropy $H$ is calculated for chosen cue as:

$$H(S_t) = -\sum_i P(S_t) log P(S_t) \tag{14}$$

## Hierarchical Bayesian model

The Hierarchical Bayesian model introduced by (*Mathys et al., 2011*) incorporates different forms of uncertainty during learning on each level: irreducible uncertainty (resulting from probabilistic relationship between prediction and outcome), estimation uncertainty (from imperfect knowledge of stimulus-outcome relationship), and volatility uncertainty (from potential environmental instability). This model has been shown to fit human acute stress responses (*de Berker et al., 2016*). The model was adopted to our study with the basic structure unchanged, and the second level estimated probabilities were used to approximate state values of different cues, and the preferred action calculated using the softmax function.

## Modelling pain ratings

Our prior hypothesis suggests uncertainty is a likely modulator of tonic pain perception, hence model generated uncertainty signals (associability in Experiments 1 and 2, with entropy and surprise added in Experiment 2) were used as the main pain rating predictors. A generalised linear model includes the uncertainty predictor, and additional terms to control for potential temporal habituation/sensitization and between-session variation:

$$\text{Rating} = \beta_1 \cdot \text{Relief} + \beta_2 \cdot log(\text{Trial}) + \beta_3 \cdot \text{Predictor} \tag{15}$$

where the 'Relief' term is the number of trials since the previous relief outcome, $log(\text{Trial})$ is the log of trial number within session (1-24), 'Predictor' is the model generated uncertainty value using group-averaged model parameters fitted with choice/SCR data. All trials were used for predictor calculation, but only rated trials were included in this regression.

## Model fitting and comparison
### Model fitting

Model fitting was performed with the Variational Bayesian Analysis (VBA) toolbox (https://mbb-team.github.io/VBA-toolbox/). The toolbox seeks to optimise free energy within the Bayesian framework, analogous of maximum likelihood. Behavioural data (choices, SCRs) were fitted separately for each individual resulting in different sets of parameters, and model fitting performance was measured by aggregating individual subject fitting statistics. The mean of all subject parameters were used to generate regressors for fMRI analysis following conventions (*Table 5* and *Table 6*).

The VBA toolbox takes in an evolution function that describes the learning model (e.g. value updating rule), and an observation function that describes response mapping (e.g. softmax action selection). For choice fitting, data were split into multiple sessions to allow between-session changes in observation function parameters, but evolution function parameters and initial states were fixed throughout all sessions.

For SCR fitting, multi-session split was the same as choice fitting. The first two trials from each session were excluded from fitting to avoid extreme values from startle effects, which also served to reduce the confound from general habituation of SCRs. Trials with insufficient event-related responses were also excluded (see 'Physiological measures' above). The observation function for

**Table 5.** Experiment 1 learning model fitting results.

| Model (Options) | Data fitted (sessions) | Parameters | Mean | Std | Initial states |
|---|---|---|---|---|---|
| TD (*) | choice (instrumental) | learning rate, $\alpha$ | 0.401 | 0.087 | $Q_0=0$ |
| WSLS (*) | choice (instrumental) | pseudo Q (cue 1), p1 | 0.382 | 0.073 | No hidden states |
| | | pseudo Q (cue 2), p2 | 0.458 | 0.075 | |
| Hybrid Action learning (*) | choice (instrumental) | free parameter $\kappa$ | 0.527 | 0.104 | $Q_0=0$ |
| | | free parameter $\eta$ | 0.413 | 0.125 | $\alpha_0=1$ |
| RW - V (†) | SCR (instrumental) | learning rate, $\alpha$ | 0.492 | 0.013 | $V_0=0$ |
| RW - V (†) | SCR (Pavlovian) | learning rate, $\alpha$ | 0.492 | 0.014 | $V_0=0$ |
| Hybrid - Assoc (†) | SCR (instrumental) | free parameter $\kappa$ | 0.497 | 0.004 | $V_0=0$ |
| | | free parameter $\eta$ | 0.495 | 0.004 | $\alpha_0=1$ |
| Hybrid - Assoc (†) | SCR (Pavlovian) | free parameter $\kappa$ | 0.498 | 0.003 | $V_0=0$ |
| | | free parameter $\eta$ | 0.496 | 0.008 | $\alpha_0=1$ |
| Hybrid - V (†) | SCR (instrumental) | free parameter $\kappa$ | 0.492 | 0.012 | $V_0=0$ |
| | | free parameter $\eta$ | 0.499 | 0.003 | $\alpha_0=1$ |
| Hybrid - V (†) | SCR (Pavlovian) | free parameter $\kappa$ | 0.494 | 0.005 | $V_0=0$ |
| | | free parameter $\eta$ | 0.5 | 0.003 | $\alpha_0=1$ |

*Fitting options: muTheta, muPhi = 0, sigmaTheta, sigmaPhi = 1.
†muTheta, muPhi=0, sigmaTheta=0.05, sigmaPhi=1.
DOI: https://doi.org/10.7554/eLife.31949.011

**Table 6.** Experiment 2 learning model fitting results.

| Model (Options) | Data fitted | Parameters | Mean | Std | Initial states |
|---|---|---|---|---|---|
| TD (*) | choice | learning rate, $\alpha$ | 0.577 | 0.28 | $Q_0=0$ |
| Hybrid Action learning (*) | choice | free parameter $\kappa$ | 0.774 | 0.381 | $Q_0=0$ |
| | | free parameter $\eta$ | 0.14 | 0.139 | $\alpha_0=1$ |
| HMM (*) | choice | state transition probability $\beta$ | 0.275 | 0.213 | $Q_0=0.5$ |
| | | relief outcome bias c | 0.535 | 0.212 | |
| | | no relief outcome bias d | 0.027 | 0.072 | |
| Bayesian (‡) | choice | level 2 (outcome) $\kappa$ | 0.331 | 0.239 | $Q_0=0$ |
| | | level 2 (outcome) $\omega$ | −0.423 | 1.396 | |
| | | level 3 (belief) $\theta$ | 0.45 | 0.03 | |
| RW - V (†) | SCR (bilateral) | learning rate, $\alpha$ | 0.46 | 0.054 | $V_0=0$ |
| Hybrid - Assoc (†) | SCR (bilateral) | free parameter $\kappa$ | 0.49 | 0.01 | $V_0=0$ |
| | | free parameter $\eta$ | 0.488 | 0.027 | $\alpha_0=1$ |
| Hybrid - V (†) | SCR (bilateral) | free parameter $\kappa$ | 0.48 | 0.034 | $V_0=0$ |
| | | free parameter $\eta$ | 0.496 | 0.013 | $\alpha_0=1$ |

* Fitting options: muTheta, muPhi = 0, sigmaTheta, sigmaPhi = 1.

†muTheta, muPhi = 0, sigmaTheta = 0.05, sigmaPhi = 1.

‡muTheta=[0,-2,0], muPhi=0, sigmaTheta, sigmaPhi=1

DOI: https://doi.org/10.7554/eLife.31949.017

SCR fitting were simply $g(x) = \text{Predictor} + \text{b}$, with $b$ as a free parameter. The predictor (model uncertainty) was not scaled to avoid overfitting. For Experiment 2, both left and right SCRs were fitted simultaneously as two data sources, with $b_1$ and $b_2$ as two free parameters to fit each side with the same predictor.

Parameter prior setting for models followed previous studies. TD, RW and hybrid models all have initial values as 0, and initial associability as 1. HMM and Bayesian models all have initial hidden states of relief belief as 0. All evolution parameters had variance set to 1, with the exception of SCR fitting at 0.05 to reduce flexibility.

We calculated the protected exceedance probabilities based on (*Rigoux et al., 2014*), shown in figure supplements in the same way as in the original exceedance probabilities in the results section. See http://mbb-team.github.io/VBA-toolbox/wiki/BMS-for-group-studies/#rfx-bms for details of its calculation. In Experiment 1, for SCR fitted model comparison, the best fitting model became less clear. However, in Experiment 2, where the number of trials was increased as fitting was not conducted separately for Instrumental/Pavlovian sessions, best fitting models from comparison remained unchanged from the original comparison using exceedance probabilities. Results from Experiment 2 provided validation for Experiment 1 in the way similar to the neuroimaging analysis.

## Model comparison

Model comparison was implemented with random-effect Bayesian model selection in the VBA toolbox. The best fitted model for each individual is allowed to vary, and model frequency in population (i.e. in how many subjects the model was the best-fit model) was estimated from model fitting evidence (free energy from learning models in choice and SCR fitting, or log likelihood from regression models in rating fitting), and model exceedance probability (i.e. how likely the model is more frequent than other models compared).

## fMRI acquisition

For Experiment 1, neuroimaging data was acquired with a 3T Siemens Magnetom Trio Tim scanner, with the Siemens standard 12 channel phased array head coil. For Experiment 2, a 3T Siemens Prisma scanner was used, with the Siemens standard 64 channel phased array head coil.

Scanning parameters were identical for both experiments: functional images were collected with a single echo EPI sequence (repetition time TR = 2500 ms, echo time TE = 30 ms, flip angle = 80, field of view = 240 mm), 37 contiguous oblique-axial slices (voxel size 3.75 × 3.75×3.75 mm) parallel to the AC-PC line were acquired. Whole-brain high resolution T1-weighted structural images (dimension 208 × 256×256, voxel size 1 × 1×1 mm) using standard MPRAGE sequence were also obtained.

## fMRI preprocessing

Functional images were slice time corrected using SPM12 (http://www.fil.ion.ucl.ac.uk/spm/software/spm12/) with individual session's slice timing output by the scanner. Resulting images were then preprocessed using the fmriprep software (build date 09/03/2017, freesurfer option turned off, https://github.com/poldracklab/fmriprep), a pipeline that performs motion correction, field unwarping, normalisation, field bias correction, and brain extraction using a various set of neuroimaging tools available. The normalised images were smoothed using a Gaussian kernel of 8 mm using SPM12. The confound files output by fmriprep include the following signals: mean global, mean white matter tissue class, three FSL-DVARS (stdDVARS, non-stdDVARS and voxel-wise stdDVARS), framewise displacement, six FSL-tCompCor, six FSL-aCompCor, and six motion parameters (matrix size: 24 × number of volumes).

## fMRI GLM model

All event-related fMRI data were analysed with generalised linear models (GLM) constructed using SPM12, estimated for each participant in the first level. Model generated signals used as parametric modulators were generated with one set of group-mean model parameters, obtained with behavioural data fitting as described. We used the mean of the fitted parameters from all participants in the imaging analysis as this provides the most stable estimate of the population mean (taking into account the fact that individual fits reflect both individual differences and noise). For completeness, however, we also ran the analyses with individually fitted values, which led to similar results (i.e. no change in significance level of each result). All regressors were convolved with a canonical hemodynamic response function (HRF). We also include regressors of no interest to account for habituation and motion effects. Specifically, the number of trials since last receiving a relief outcome ('Relief' term in rating regression model), and the log of trial number within session (log(Trial) term) were included to regress out potential change in tonic pain perception simply due to prolonged stimulation. The resulting GLM estimates were entered into a second-level one-sample t-test for the regressors of interest to produce the random-effect statistics and images presented in Results section.

### TD softmax (*Figure 3a* and *Figure 5a*)

Regressors of interest:

- CS onset (duration = 3 s, cue presentation): $Q$ values of chosen cue,
- Outcome onset (duration = 3 s): prediction error,

Regressors of no interest:

1. CS onset (duration = 3 s, cue presentation): number of trials since last relief,
2. CS onset (duration = CS onset to outcome offset, entire trial exclude ITI): within session log trial number,
3. choice press (duration = 0),
4. rating press (duration = rating duration),
5. CS offset (duration = 0),
6. 24 column confounds matrix output by fmriprep.

### Hybrid model associability (*Figure 3d* and *Figure 5d*)

Regressors of interest:

- choice press time (duration = 0, cue button press): associability (generated for individual session with new $V_0/A_0$ to match SCR fitting procedure),

Regressors of no interest: same as GLM above, adding relief onset (duration = 0), and removing choice press regressor. We note for completeness that it is theoretically possible to model the learning process as a continuously valued function that exactly matches the time-course of the temperature changes. In the context of the current study, this effect of this would be largely orthogonal to the experimental manipulations. However, representation of the baseline temperature as a continuous function is clearly important in real-life contexts in which the baseline level determines homeostatic motivation and phasic reward functions (*Morville et al., 2018*), and hence future studies could directly manipulate this.

For multiple comparison, we used anatomical binary masks generated using the Harvard-Oxford Atlas (*Desikan et al., 2006*) for small volume correction. Atlases are freely available with the FSL software (https://fsl.fmrib.ox.ac.uk/fsl/fslwiki/Atlases). We thresholded the probability maps at 50%, focusing on ROIs defined a priori (learning related: amygdala, accumbens, putamen, caudate, pallidum, VMPFC, DLPFC. Controllability-induced analgesia related: cingulate gyrus - anterior division, insular cortex, VLPFC). We used the frontal medial cortex for VMPFC, the frontal orbital cortex for VLPFC, and the middle frontal gyrus for DLPFC respectively. We reported results with $p < 0.05$ (FWE cluster-level corrected). Masks were applied separately, not combined (*Table 1* and *Table 2*).

## fMRI model comparison

To determine whether state-based and action-based learning involve the same brain regions during instrumental learning, we used Bayesian model selection (BMS) with the instrumental sessions imaging data. We ran Bayesian first level analysis using two separate GLMs containing the prediction error signals from TD and hybrid models (at outcome onset time, durations = 3 s) using unsmoothed functional imaging data, with the same regressors of no interest as other GLMs described. To reduce computation time, this was restricted to voxels correlated to prediction error from previous parametric modulation analysis results from our present study, within a mask of conjunction clusters from TD and hybrid prediction error analysis (cluster formation at $p < 0.01$, $k < 5$). Resulting log-model evidence maps produced from each model for individual participant were first smoothed with a 6 mm Gaussian kernel, then entered into a random-effect group analysis (*Stephan et al., 2009*). Voxel-wise comparison between models produced posterior and exceedance probability maps to show whether a particular brain region is better accounted for by one model or the other. Posterior probability maps were overlaid on subject-averaged anatomical scans using MRIcroGL (https://www.nitrc.org/projects/mricrogl/).

## Axiom analysis for prediction errors

To determine whether ROI activations to prediction errors were responding outcomes or prediction errors, we carried out ROI axiomatic analysis (*Roy et al., 2014*). Trials were separated into relief or no relief outcomes, then into equal-size bins of ascending sorted expected relief values, calculated from TD model as we were primarily interested in instrumental/active relief learning. This produced four regressors (2 outcomes × 2 value bins) in Experiment 1, and six regressors (2 outcomes × 3 value bins) in Experiment 2, to be estimated at outcome time (duration = 3 s) when prediction error was generated. GLMs include button presses for choice or rating, and movement related regressors of no interest mentioned above. ROI masks of 8 mm spheres were generated from peak coordinates from TD model prediction error exceedance probability map calculated by BMS above (ventral and dorsal striatum, amygdala, VMPFC and DLPFC).

Mean activity were extracted from these ROI masks averaged across sessions within individual subject. Although the axiomatic analysis is useful for delineating outcome and prediction responses in previous reward or aversive PE studies, the continued presence of tonic pain in our study differs from the 'no stimulation' conditions in these studies, thus we are primarily interested in the overall BOLD activity pattern and did not include full statistics of this analysis.

## Acknowledgements

Research was supported by National Institute for Information and Communications Technology (Japan), the Wellcome Trust (UK, Ref: 097490), the Japanese Society for the Promotion of Science (JSPS), and the 'Application of DecNef for development of diagnostic and cure system for mental disorders and construction of clinical application bases' of the Strategic Research Program for Brain

Sciences from Japan Agency for Medical Research and development, AMED. SZ is supported by the WD Armstrong Fund and the Cambridge Trust. The Research is also supported by Arthritis Research UK (Ref: 21357), We thank Drs. Daniel McNamee and Agnes Norbury for helpful discussions, and the imaging teams at the Center for Information and Neural Networks and the Advanced Telecommunications Research Institute for their assistance in performing the study.

## Additional information

### Funding

| Funder | Grant reference number | Author |
|---|---|---|
| National Institute of Information and Communications Technology | | Suyi Zhang<br>Hiroaki Mano<br>Ben Seymour |
| Cambridge Commonwealth Trust | | Suyi Zhang |
| Japan Society for the Promotion of Science | S2604 | Hiroaki Mano<br>Wako Yoshida<br>Ben Seymour |
| Japan Agency for Medical Research and Development | | Wako Yoshida<br>Mitsuo Kawato<br>Ben Seymour |
| Wellcome Trust | 097490 | Trevor W Robbins<br>Ben Seymour |
| Arthritis Research UK | 21357 | Ben Seymour |
| WD Armstrong Fund | | Suyi Zhang |

The funders had no role in study design, data collection and interpretation, or the decision to submit the work for publication.

### Author contributions

Suyi Zhang, Conceptualization, Resources, Data curation, Software, Formal analysis, Validation, Investigation, Visualization, Methodology, Writing—original draft, Writing—review and editing; Hiroaki Mano, Conceptualization, Validation, Methodology, Writing—original draft, Writing—review and editing; Michael Lee, Wako Yoshida, Mitsuo Kawato, Trevor W Robbins, Conceptualization, Methodology, Writing—original draft, Writing—review and editing; Ben Seymour, Conceptualization, Data curation, Formal analysis, Supervision, Funding acquisition, Validation, Investigation, Visualization, Methodology, Writing—original draft, Writing—review and editing

### Author ORCIDs

Suyi Zhang  http://orcid.org/0000-0001-9028-6265
Wako Yoshida  http://orcid.org/0000-0001-9273-1617
Ben Seymour  http://orcid.org/0000-0003-1724-5832

### Ethics

Human subjects: The two experiments were performed in different institutes, and approved by their relevant ethics and Safety committees: for the National Institute of Information and Communications Technology, Japan (Expt 1), and the Advanced Telecommunications Research Institute, Japan (Expt 2). All subjects gave informed consent prior to participation

### Decision letter and Author response

Decision letter https://doi.org/10.7554/eLife.31949.028
Author response https://doi.org/10.7554/eLife.31949.029

## Additional files

### Supplementary files

• Transparent reporting form
DOI: https://doi.org/10.7554/eLife.31949.024

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
