## [Decision Letter]

Thank you for submitting your article "The Control of Tonic Pain by Active Relief Learning" for consideration by *eLife*. Your article has been reviewed by three peer reviewers, and the evaluation has been overseen by a Reviewing Editor and Michael Frank as the Senior Editor. The reviewers have opted to remain anonymous.

The reviewers have discussed the reviews with one another and the Reviewing Editor has drafted this decision to help you prepare a revised submission.

Summary:

Zhang and colleagues investigate how different aspects of relief learning during tonic pain stimulation relate to pain perception and what the neuronal correlates of these processes are. They report that uncertainty/attention inferred from a formal model parameter called 'associability' is correlated to reductions in pain perception. This is an intriguing and novel take on the links between learning processes and pain regulation. The study addresses an important and timely question that will be of high interest to readers in various research fields, including pain, learning theory, decision-making, and motivation. They identify neural correlates of prediction error and associability and show that these parameters map onto responses in striatum and pgACC in two separate imaging studies of instrumental relief learning during tonic heat pain (effects during Pavlovian relief learning are less conclusive). The paper is well-written, analyses are appropriate, and the work builds on previous studies of tonic pain and pain-related learning from this group, as well as a growing body of work integrating pain and associative learning.

Essential revisions:

All reviewers noted that the modeling was sophisticated but not particularly accessible to a non-modeling audience. Overall, the manuscript is densely written and relies a lot on technical terms. Unpacking some of the ideas and concepts in the Introduction and Results section would help to make the manuscript more accessible to a broader audience. Given that this is a general interest journal, I hope the following suggestions will make it easier to for a broader audience to extract conclusions. At the same time there are technical concerns that need to be addressed. These comments reflect input from all three reviewers – there are many points but some of them are convergent.

1) It isn't clear why the particular models tested were chosen and what adjudicating between them tells us, in practical terms. For example, what are the implications of a model with a fixed learning rate (TD) fitting better than a model with an adaptive learning rate (hybrid TD)? Also, as it seems a central goal to demonstrate that RL models have more explanatory power than simpler models, it would be helpful to be able to understand how well each model fit and what the incremental difference between them is. The latter might be accomplished by expanding on the description of exceedance probability in “subsection “Model fitting and comparison” and mentioning it in the Results (or figure captions?). Relatedly, the behavioral choice data in Experiments 1 and 2 are best explained by a temporal-difference (TD) model without an associability term. Are decisions and actions thus independent of the associability? Do participants learn, but do not act on that knowledge? What does that imply for the conclusions drawn here?

The authors fit models to individuals' behavior, then used mean parameters to generate regressors for neuroimaging data. Individuals seem quite variable in terms of fitted parameters, particularly in Experiment 2, and this variability in learning and performance might contribute to inconsistencies in the neural data. Why did the authors not (1) use the individual model fits in the imaging analyses, (2) fit to the group, or (3) incorporate information about individual fits (e.g. learning rates in the TD model) at the subject level in analyses?

Also, regarding the modeling efforts – How reliable are the model parameters and outcomes of the model comparisons, when only 16-18 SCR data points are used for fairly complex TD or Hybrid models for each session? Are the reported exceedance probabilities for the model comparisons 'protected exceedance probabilities' (Rigoux et al., 2014) that account for the possibility that models are equally (un)likely?

2) Associability is clearly a critical construct here, but it seems to arise operationally from the models but a little muddier at the level of theory. For example, would an associability account of relief learning differ from an attentional account? Associability and attention are discussed almost interchangeably. On a more practical level, simply stating the direction of associability (e.g. high associability = higher uncertainty) clearly would make correlations more immediately interpretable. Clarifying the relationship between these concepts early on and consistently using them throughout the manuscript will increase readability.

As a more general point relating to both of the preceding issues, I think some of the difficulties in interpreting results was due to the format whereby results are presented before methods. As an example, there was more description of associability in the discussion and methods, but it would have been helpful to have some information provided in the results, given that this is read first. Perhaps the authors could be mindful of this format and provide some explanation along with the results presented?

3) Other than the fact that a reinforcement learning paradigm fits putamen responses, do we have evidence that dorsal putamen responses are involved in a learning process? Is there any correspondence between dorsal putamen findings and behavioural findings?

4) The timing of pain and relief ratings wasn't very clear. Am I correct in inferring that both pain and relief ratings were collected at three time points in each session (near beginning, near middle, near end)? How many of each rating makes up the scores reported?

5) The important event for participants is the reduction of pain when the temperature is reduced. Since pain has a continuous intensity dimension and a reduction along this dimension is driving the learning process, I wonder whether the RL models could be extended to use a continuous outcome that might offer more information?

6) Given that correspondence between the two experiments is an important feature, a figure in which activations in one experiment are overlayed on the other (to judge spatial correspondence) would be helpful.

7) It isn't very clear how imaging data were corrected for multiple comparisons. Relatedly, in some cases, searches were restricted to a priori ROIs (e.g. pgACC, posterior insula, vlPFC), but isn't clear how these were defined (e.g. anatomically? Based on previous findings?) or whether data in these analyses was corrected across the mask of all ROIs. From the tables and Results section, I conclude that the authors use a mix of Cluster-extend thresholding and peak-voxel SVC correction. The authors should choose one method and use it consistently.

Furthermore, the authors use SVC correction based on coordinates from hand-selected previous studies or selectively use Experiment 1 coordinates for SVC correction of the amygdala results in Experiment 2. With the availability of comprehensive anatomical atlases, I urge the authors to apply masks based on anatomical atlases or independent functional localizers to correct for multiple comparisons.

8) Since the conclusions of this manuscript rely primarily on the model efforts, I think presenting about absolute model fits for choices and SCR data would help in evaluating the models. In addition, presenting information on SCR data quality will help to convince the reader about the conclusions. For example, skin conductance shows spontaneous, phasic responses during acute (10-20s) or tonic pain stimuli. To which degree are the responses modeled here locked to the cue- or outcome-events? Showing raw SCR traces and/or averaged evoked responses with predicted SCR responses would help here, e.g. using *eLife*'s figure supplements.

9) Do changes in pain perception also correlate with pgACC activity when used as a regressor in subject-level models?

10) In the Discussion section, the authors argue that lack of controllability in Pavlovian paradigms renders uncertainty hyperalgesic instead of analgesic. However, pain ratings do not differ between instrumental and Pavlovian sessions in Experiment 1, as predicted by this reasoning.

11) In subsection “Ratings” the authors argue that placebo expectation theory predicts that larger prediction errors are correlated with pain reductions. Montgomery & Kirsch, (1997) and Locher et al., (2017) have shown that a plausible instruction regarding the placebo is needed for conditioned placebo analgesia. Participants in the present study weren't given any rationale for a cue being a placebo treatment. Hence, different processes might be involved here. In addition, this test relies on the correct estimation of the prediction error, which depends on the estimated value. The value will increase (i.e. encode more expectation for relief) over repeated trials that included a relief. When the expectation for relief and thus the prediction errors are maximal, participants have just experienced a series of relief trials and the surprise or associability/uncertainty to previous trials is also maximal.

12) In both studies, pain (and relief) ratings were collected "intermittently," yet the authors make strong assumptions about effects on relief/pain based on the correlations between ratings and the time-varying measures of associability or prediction error. The authors should present complete information about rating measurement for each experiment (e.g. number of ratings) and justify why they did not incorporate ratings at the same time scale of choice, stimulus display, and SCR measurement. To determine that ratings are preferentially related to associability and not prediction error, it seems that all quantities should be measured with the same number of observations. Furthermore, this would allow direct fits to ratings, which would be the best way to determine how these learning-related parameters modulate pain and relief. Finally, if I understand correctly, Experiment 2 included pain ratings before relief outcomes were delivered. These ratings are likely to be influenced by anticipation and uncertainty, but not by relief, whereas Experiment 1's ratings were measured after outcomes. Thus, the studies differ in terms of the construct that is captured by ratings. Since pain and relief are ultimately subjective, a more thorough consideration of the self-report measures is warranted.

13) Please also explain the negative coefficients in Figure 4E – participants experienced less pain with higher associability and with longer time since relief? This seems inconsistent with previous work on uncertainty, attention, and desire for relief which should enhance pain.

14) Skin conductance was found to be best fit by associability from the hybrid model. Can the authors rule out the possibility that this is only the case because both associability and skin conductance decrease over time? Other models included an effect of time/trial to account for such habituation. Are these findings artefactual, and might SCR track value or prediction error if habituation is modeled separately?

15) The study uses a mild tonic stimulus in healthy volunteers and measures behavioral correlates of intermittent relief. While the pgACC results are cool, I find it quite inappropriate to suggest that "the results highlight the pgACC as a target for therapeutic intervention […] by invasive excitatory deep brain stimulation."

[Editors' note: further revisions were requested prior to acceptance, as described below.]

Thank you for resubmitting your work entitled "The Control of Tonic Pain by Active Relief Learning" for further consideration at *eLife*. Your revised article has been favorably evaluated by Michael Frank (Senior editor), a Reviewing editor, and three reviewers.

The manuscript has been improved but there are some remaining issues that need to be addressed before acceptance, as outlined below:

*Reviewer #1:*

The authors have largely satisfied any concerns I had about the reliability of the findings. So, I would be comfortable publishing the paper in its current form.

That said, I concur that they haven't done as much as they might have to increase the paper's accessibility to a general audience. On re-reading the paper after the authors' responses, I think the easiest way to do this might be to do some additional revision to the introduction. The Introduction (particularly before the addition of the sections on associability and reward learning) does little to set up the actual experimental paradigms and modelling techniques used, such that one ends up trying to piece together the rationale for most of what was done while reading the methods and results. The methodology and modelling would have been far clearer to me had the authors been more explicit (as they were in their reply to reviewers) about the relevance of associability for illuminating the distinction between state and action learning and how doing so relates to the broader goal of understanding relief learning in the context of tonic pain.

So, in summary, the paper is publishable, but I do think the paper could be improved in terms of accessibility without a great deal of additional work.

Reviewer #2:

The authors have addressed all my comments and questions.

Reviewer #3:

For the most part, the authors have addressed all major concerns. I was particularly impressed that results and conclusions hold (1) whether parametric modulators are based on individual versus mean fits for Experiment 2 (although I think the authors should consider including these results in Supplementary figures), and (2) when consistently defined ROIs are employed (new Tables 6 and 7). The paper is also strengthened by the addition of information clarifying rating procedures and depicting skin conductance over time.

However, I feel that a few concerns remain, which I have delineated as minor concerns in the following section. In several places (e.g. the discussions of contingency awareness, modeling the time course of temperature changes, subjectivity of ratings), I felt that the authors only superficially engaged with reviewers' collective suggestions, and that overall accessibility of the work is still somewhat limited for non-expert audiences.

---

## [Author Response]

Essential revisions:All reviewers noted that the modeling was sophisticated but not particularly accessible to a non-modeling audience. Overall, the manuscript is densely written and relies a lot on technical terms. Unpacking some of the ideas and concepts in the Introduction and Results section would help to make the manuscript more accessible to a broader audience. Given that this is a general interest journal, I hope the following suggestions will make it easier to for a broader audience to extract conclusions. At the same time there are technical concerns that need to be addressed. These comments reflect input from all three reviewers – there are many points but some of them are convergent.

We thank the reviewers for pointing out the issue with accessibility for non-modelling audience. We have modified the manuscript to improve this, and these changes are detailed in response to the specific issues below. But briefly, this involves (i) explaining the modelling aspects in a more intuitive manner, and (ii) moving some model description contents from The Discussion section and Materials and methods section to the Introduction and Results section.

1) It isn't clear why the particular models tested were chosen and what adjudicating between them tells us, in practical terms. For example, what are the implications of a model with a fixed learning rate (TD) fitting better than a model with an adaptive learning rate (hybrid TD)? Also, as it seems a central goal to demonstrate that RL models have more explanatory power than simpler models, it would be helpful to be able to understand how well each model fit and what the incremental difference between them is. The latter might be accomplished by expanding on the description of exceedance probability on page 19 and mentioning it in the Results (or figure captions?). Relatedly, the behavioral choice data in Experiments 1 and 2 are best explained by a temporal-difference (TD) model without an associability term. Are decisions and actions thus independent of the associability? Do participants learn, but do not act on that knowledge? What does that imply for the conclusions drawn here?

There are a few issues here for us to clarify in the manuscript:

i) Choice of RL models: The models chosen for comparison were RL models that have been well-evidenced in previous reward learning studies. In many ways, RL/TD models are the simplest mechanistic models of learning and reflect an account of learning that is both intuitive and in particular, well-grounded in the animal learning theory literature. We now include an additional sentence in Introduction to set this out with much more clarity:

“RL models provide a mechanistic (as opposed to merely descriptive) account of the information processing operations that the brain actually implements and have a solid foundation in classical theories of animal learning (Mackintosh, 1983; Dayan and Abbott, 2001).”

ii) The nature and role of associability. The behavioural choice data are best fitted by TD model without an associability term, while SCR data are best fitted by hybrid TD model with an associability term. This difference between action and state learning has been observed in previous studies (Li et al., 2011; Boll et al., 2013; Zhang et al., 2016; Gläscher et al., 2010; Morris et al., 2006), and reflects the distinction between instrumental and Pavlovian learning systems. The key point is that Pavlovian and instrumental systems learn different things – conditioned responses (such as autonomic responses) and actions respectively, each of which have independent biological functions. So, whereas we do not find evidence that associability is used in determining actions (in keeping with previous reports), it *is* used for learning conditioned responses. Although these are independent in our experiment and model, it is perfectly possible (even, likely) that they do interact in appropriate circumstances, but we haven’t employed a task to probe this here. We now mention this more explicitly in the manuscript, in the Introduction:

“Models of the role of attention during learning typically invoke attention during uncertainty, as this is when there may be the greater requirement to devote resources to enhance learning. From a computational perspective, the role of uncertainty is often operationalised as controlling the learning rate, such that high uncertainty (hence high attention) leads to more rapid learning (Dayan et al., 2000; Angela and Dayan, 2005). One way of formalising uncertainty in RL by computing a quantity called the associability, which calculates the running average of the magnitude of recent prediction errors (i.e. frequent large prediction errors implies high uncertainty / associability). The concept of associability is well grounded in classical theories of Pavlovian conditioning (the `Pearce-Hall' learning rule, Pearce and Hall, 1980; Le Pelley, 2004; Holland and Schiffino, 2016), and provides a good account of behaviour and neural responses during Pavlovian learning (Li et al., 2011; Boll et al., 2013; Zhang et al., 2016). In this way, it can be seen that associability reflects a computational construct that captures aspects of the psychological construct of attention.”

Later in the Results section, we emphasize this again;

“As mentioned above, the associability reflects the uncertainty in the action value, where higher associability indicates high uncertainty during learning, and is calculated based the recent average of the prediction error magnitude for each action.”

And again, point out the fact that associability does not to state-learning only (Results section).

“Thus, there is no evidence that associability operates directly at the level of actions”.

And later in subsection “Ratings”:

“This divergence in learning strategies indicates that parallel learning systems coexist, which differ in their way of incorporating information about uncertainty in learning, as well as the nature of their behavioural responses.”

iii) Explaining exceedance probability: Since we had a specific prior hypothesis of a reciprocal relationship between attention and tonic pain, we used hybrid TD model to test this hypothesis by demonstrating that it fitted better than conventional RL models in relief learning. We have incorporated more information on model exceedance probability in Figure 2 legend to familiarise audience with the concept:

“Model frequency represents how likely a model generate the data given a random participant, while exceedance probability estimates how one model is more likely compared to others (Stephen et al., 2009).”

The authors fit models to individuals' behavior, then used mean parameters to generate regressors for neuroimaging data. Individuals seem quite variable in terms of fitted parameters, particularly in Experiment 2, and this variability in learning and performance might contribute to inconsistencies in the neural data. Why did the authors not (1) use the individual model fits in the imaging analyses, (2) fit to the group, or (3) incorporate information about individual fits (e.g. learning rates in the TD model) at the subject level in analyses?

We conducted the additional analysis as the reviewer suggested, using individual behavioural data fitted parameters to produce sequences of parametric modulators, instead of using a single group-mean parameter. Author response image 1 showed overlayed clusters from TD model prediction errors with group mean learning rate in Experiment 2 (red), and prediction errors with individual learning rates (green). Resulting dark green clusters were the overlapping regions between the two maps (both viewed at p<0.001 unc). Using individual learning rate reproduced the major clusters reported in the manuscript (dorsal putamen, amygdala, middle frontal gyrus left), with the global peak in left putamen ([-32, -12, 5], T=5.89, where group-mean parameter analysis had the global peak in the same coordinate with T=6.98). Given that Experiment 2 TD learning rates had the largest variation comparing to all other fitted models (mean=0.577, with standard deviation=0.28, see Tables of model fitting results), it’s reasonable to believe that other model results are not likely to change significantly when using individual instead of group-mean parameters.

We now mention this result briefly in the manuscript, in the appropriate section of the Materials and methods section, although we don’t include the figure for the sake of clarity but present it here to reassure the reviewers of the validity of the point.

“We used the mean of the fitted parameters from all participants in the imaging analysis as this provides the most stable estimate of the population mean (taking into account the fact that individual fits reflect both individual differences and noise). For completeness, however, we also ran the analyses with individually fitted values, which led to similar results (i.e. no change in significance level of each result).”

Also, regarding the modeling efforts – How reliable are the model parameters and outcomes of the model comparisons, when only 16-18 SCR data points are used for fairly complex TD or Hybrid models for each session? Are the reported exceedance probabilities for the model comparisons 'protected exceedance probabilities' (Rigoux et al., 2014) that account for the possibility that models are equally (un)likely?

To clarify this point, we note that we used SCR data points from all sessions for model fitting, applying multi-session split within VBA toolbox, which adjusted fitting considering all session data (Experiment 1: 3.3 sessions/~60 trials per participant, Experiment 2: 4.8 session/~115 trials per participant). Also, we constrained free parameters to have a low variance to limit overfitting in order to allow model generalisability.

Notwithstanding this, however, we have followed the reviewer’s advice to calculate the protected exceedance probabilities. We now add the following clarification in the Materials and methods section, and the additional results in Figure 2—figure supplement 3 and Figure 4—figure supplement 3:

“We calculated the protected exceedance probabilities based on Rigoux et al., (2014), shown in figures supplements in the same way as in the original exceedance probabilities in the Results section. See http://mbb-team.github.io/VBA-toolbox/wiki/BMS-for-group-studies/#rfx-bms for details of its calculation. In Experiment 1, for SCR fitted model comparison, the best fitting model became less clear. However, in Experiment 2, where the number of trials was increased as fitting wasn’t conducted separately for Instrumental/Pavlovian sessions, best fitting models from comparison remained unchanged from the original comparison using exceedance probabilities. Results from Experiment 2 provided validation for Experiment 1 in the way similar to the neuroimaging analysis.”

2) Associability is clearly a critical construct here, but it seems to arise operationally from the models but a little muddier at the level of theory. For example, would an associability account of relief learning differ from an attentional account? Associability and attention are discussed almost interchangeably. On a more practical level, simply stating the direction of associability (e.g. high associability = higher uncertainty) clearly would make correlations more immediately interpretable. Clarifying the relationship between these concepts early on and consistently using them throughout the manuscript will increase readability.

Clearly there is a strong parallel between associability (a computational construct) and attention (a psychological construct), such that whilst they cannot be directly equated (as they are different types of construct), it is certainly the case that they share a common intuitive basis. We have modified the text in manuscript to discuss associability in terms of uncertainty, and this includes the major new introductory paragraph as mentioned above. We hope this makes the narrative more consistent.

See above.

As a more general point relating to both of the preceding issues, I think some of the difficulties in interpreting results was due to the format whereby results are presented before methods. As an example, there was more description of associability in the discussion and methods, but it would have been helpful to have some information provided in the results, given that this is read first. Perhaps the authors could be mindful of this format and provide some explanation along with the results presented?

Again, this point echoes the preceding points, which we have dealt with above.

3) Other than the fact that a reinforcement learning paradigm fits putamen responses, do we have evidence that dorsal putamen responses are involved in a learning process? Is there any correspondence between dorsal putamen findings and behavioural findings?

It is hard for us to provide causative evidence of a role of dorsal putamen in the actual learning process itself, and the difficulty in distinguishing the site of learning from performance has been a long-standing debate in resolving sub-regions on the striatum. Clearly the BOLD signal observed in our and other studies *reflects* the computation of a prediction error, and this prediction error is used in learning, but in the absence of interventional studies, we cannot conclude more than this.

4) The timing of pain and relief ratings wasn't very clear. Am I correct in inferring that both pain and relief ratings were collected at three time points in each session (near beginning, near middle, near end)? How many of each rating makes up the scores reported?

For Experiment 1, both pain and relief ratings were collected at 3 time points in each session (near beginning, near middle, near end), when there were pain / relief outcomes respectively. The results reported in Figure 2C consisted of 19 subject’s ratings averaged across sessions, separated according to paradigm (instrumental/Pavlovian) and outcome (relief/pain). For Experiment 2, only pain rated were collected in 10 random trials out of 24, the same for each session. We have included more details of rating in both the figure legends and the manuscript text and added Table 1.

5) The important event for participants is the reduction of pain when the temperature is reduced. Since pain has a continuous intensity dimension and a reduction along this dimension is driving the learning process, I wonder whether the RL models could be extended to use a continuous outcome that might offer more information?

This is an interesting point, and theoretically it is possible to model learning with continuous time and value functions. However, in our experiment, it would simply render the analyses much more complicated i.e. involve inclusion of several additional parameters, whilst being not easy to see how it would change our ability to answer the central questions being asked. To be more specific, the cues would still elicit a value signal that reflects the accumulated anticipated value, but this value would be a more complex integral of a temporal function; and the outcome signal in particular would be complicated by the fact that it’s onset was less temporally discrete. In RL generally, this latter issue is certainly important and not particularly well studied, but here, not capturing this full complexity does not confound our analysis, even if we knew how to parameterise. In this case, it might make a very subtle improvement in the sensitivity of analysis, but the findings are already sufficiently robust. Therefore, in our current analysis there is no real need *not to* treat relief learning as driven by discrete relief events (we note also our short trial design, where cue and choice period lasting for 3 seconds, and outcome events lasting for 3-4 seconds, comparing with the relatively long TR of 2.5s in fMRI acquisition). We now add the following sentence on this point in the Materials and methods section:

“We note that it is theoretically possible to model the learning process as a continuously valued function that exactly matches the time-course of the temperature changes, but such models are unnecessarily complex and largely orthogonal to the experimental manipulations.”

6) Given that correspondence between the two experiments is an important feature, a figure in which activations in one experiment are overlayed on the other (to judge spatial correspondence) would be helpful.

We have added a figure showing the overlaid associability-correlated pgACC responses (left), and prediction error correlated activations in dorsal putamen and amygdala (right), as Figure 5—figure supplements 1 and 2 in the manuscript.

7) It isn't very clear how imaging data were corrected for multiple comparisons. Relatedly, in some cases, searches were restricted to a priori ROIs (e.g. pgACC, posterior insula, vlPFC), but isn't clear how these were defined (e.g. anatomically? Based on previous findings?) or whether data in these analyses was corrected across the mask of all ROIs. From the tables and Results section, I conclude that the authors use a mix of Cluster-extend thresholding and peak-voxel SVC correction. The authors should choose one method and use it consistently.Furthermore, the authors use SVC correction based on coordinates from hand-selected previous studies or selectively use Experiment 1 coordinates for SVC correction of the amygdala results in Experiment 2. With the availability of comprehensive anatomical atlases, I urge the authors to apply masks based on anatomical atlases or independent functional localizers to correct for multiple comparisons.

Following the reviewer’s advice, we have now consistently used binary masks generated from an anatomical atlas for small volume correction. ROI masks were generated using sing the Harvard-Oxford Atlas (Desikan et al., 2006), freely available with the FSL software (https://fsl.fmrib.ox.ac.uk/fsl/fslwiki/Atlases). We thresholded the probability maps at 50%, focusing on ROIs defined a priori (learning related: amygdala, accumbens, putamen, caudate, pallidum, VMPFC, DLPFC. Controllability-induced analgesia related: cingulate gyrus – anterior division, insular cortex, VLPFC). We used the frontal medial cortex for VMPFC, the frontal orbital cortex for VLPFC, and the middle frontal gyrus for DLPFC respectively. We reported results with p<0.05 (FWE cluster-level corrected). Masks were applied separately, not combined.

In brief, there is no change in the significance levels / inference based on using a consistent system for multiple comparisons. The manuscript now includes the paragraph above and Table 3 and 7.

8) Since the conclusions of this manuscript rely primarily on the model efforts, I think presenting about absolute model fits for choices and SCR data would help in evaluating the models. In addition, presenting information on SCR data quality will help to convince the reader about the conclusions. For example, skin conductance shows spontaneous, phasic responses during acute (10-20s) or tonic pain stimuli. To which degree are the responses modeled here locked to the cue- or outcome-events? Showing raw SCR traces and/or averaged evoked responses with predicted SCR responses would help here, e.g. using eLife's figure supplements.

We have modified Table 4, to include each model’s absolute fit, as output by the VBA toolbox. The Free Energy F is the model log evidence from variational Bayesian approximation, where a larger value suggests a better fit (similar to log likelihood in conventional gradient descent fitting). While absolute model fits provide information for evaluation, calculating model frequency and model exceedance probabilities (protected) through sampling are widely accepted methods for model comparison. We have verified the results further by calculating the protected exceedance probabilities, as indicated above.

We have also added both raw SCR traces from all participants (after exclusion) and averaged filtered trial SCR traces from both experiments (Figure 2—figure supplements 1–3 and figure 4—figure supplements 1–3). Raw SCR traces showed that non-excluded participants had reliable event-evoked response within session (showing only one session from all participants without filtering, some responses might not be obvious without scaling). Trial averaged SCRs showed time locked responses to cue display, where response onsets began at ~2seconds after cue appearance, before any other events (i.e. outcome or rating) taking place. In addition, the PsPM toolbox estimates SCR by convolving skin conductance response function (SCRF) with input signal time series, allowing variable onset time and response duration, providing more accurate estimates than simple peak-to-peak measures (http://pspm.sourceforge.net/). SCR recordings from the two experiments appeared to satisfy the requirements for model fitting.

Accordingly, we have added explanation of this in the manuscript and new figures are as follows: Figure 2—figure supplement 1, Figure 2—figure supplement 2, Figure 4—figure supplement 1 and Figure 4—figure supplement 2.

9) Do changes in pain perception also correlate with pgACC activity when used as a regressor in subject-level models?

We didn’t find pain ratings correlated pgACC activations at a threshold of p<0.01 uncorrected, in keeping with the notion that associability correlated pgACC activations were not solely driven by pain perception. We conducted the analysis using pain ratings as parametric modulators at decision time (duration=0) instead of associability, as the reviewer suggested. Since Experiment 1 there were insufficient number of ratings (only 8 pain ratings per participant per paradigm), we used data from Experiment 2 only (71 ratings per participant). Two participants were excluded because of constant ratings within session, whose first-level contrasts cannot be estimated (final n=21). Clusters positively correlated with pain perception include the insula, IFG, Rolandic Operculum, while negative clusters include the precentral, primary motor cortex, the PAG. Whole-brain analysis and AAL labels were summarised in the table below (table not shown in manuscript because results were not significant).

We added the following sentence in the Results section to incorporate this result:

“In addition, we used trial-by-trial pain ratings as a parametric modulator, but did not find significant pgACC responses, which suggested that it was unlikely to be solely driven by pain perception itself.”

Experiment 2 with an initial cluster-forming threshold of p < 0.005, cluster size k>5, regions from AAL2p*kTZMNI coordinates (mm)Region (AAL)xyzPositive0.474364.373.62-36335Frontal Inf Tri L3.743.22-4078Insula L3.352.95-29188Insula L0.782244.293.5743-2023Rolandic Oper R0.986123.993.38-172246Frontal Sup 2 L0.99883.783.25-40-16-14Hippocampus L0.986123.733.2117-461Lingual R0.971143.723.254-3112Temporal Sup R3.172.8262-3512Temporal Sup R0.991113.663.1662-461Temporal Mid R0.995103.513.065-81Thalamus R0.99973.242.875-31-52Cerebelum 9 R153.232.86-407-3Insula LNegative0.99884.223.53-51-150Precentral L0.1685643.39-17-1668Precentral L3.633.15-25-1265Precentral L3.583.11-36-2065Precentral L0.991113.953.36-2-23-3PAG/Thalamus L0.99793.853.29-29-6161Parietal Sup L153.743.22-14-911Occipital Sup L0.99793.362.9532-8031Occipital Mid R3.052.7328-7638Occipital Sup R153.352.952018-3Putamen R0.99973.232.8613-6538Precuneus R

10) In the Discussion section, the authors argue that lack of controllability in Pavlovian paradigms renders uncertainty hyperalgesic instead of analgesic. However, pain ratings do not differ between instrumental and Pavlovian sessions in Experiment 1, as predicted by this reasoning.

We agree with this point. Previously we were trying to point out previous evidence where uncertainty is linked to hyperalgesia when no control is available but didn’t rule out other possibilities. But it is difficult to draw any strong conclusions, so we have now removed this sentence.

11) In subsection “Ratings” the authors argue that placebo expectation theory predicts that larger prediction errors are correlated with pain reductions. Montgomery & Kirsch (1997) and Locher et al., (2017) have shown that a plausible instruction regarding the placebo is needed for conditioned placebo analgesia. Participants in the present study weren't given any rationale for a cue being a placebo treatment. Hence, different processes might be involved here. In addition, this test relies on the correct estimation of the prediction error, which depends on the estimated value. The value will increase (i.e. encode more expectation for relief) over repeated trials that included a relief. When the expectation for relief and thus the prediction errors are maximal, participants have just experienced a series of relief trials and the surprise or associability/uncertainty to previous trials is also maximal.

We accept there is a debate about the importance of instructed or conscious contingency knowledge in generating placebo analgesic responses. In our case, we did not get explicit contingency awareness ratings, so we do not know whether they were acquired or not. The point we are trying to make is that a certain direction of response would be consistent with a placebo analgesic response, but the fact that it doesn’t occur renders the point somewhat academic. Of course, we did not design the study to actively try and pit placebo analgesia against the associability analgesic effects, although it is an interesting issue to consider. So as the reviewer correctly points out, expectation/value of relief increases after a series of relief trials, making subsequent relief outcomes less surprising, which lead to low associability/uncertainty and were associated with higher pain. And this is not the manifestation of the typical conditioned placebo analgesia, which should associate increasing expected relief values with decreased pain. Instead, therefore, the results are consistent with an information driven, temporally dynamic process that suppresses pain when learning was most needed. We cannot rule out the possibility that this is the result of the interplay between different processes, including placebo analgesia, but we can say that any placebo-like response does not dominate the ratings.

To make this clearer, we now add the following sentence in the Results section:

“… although the extent to which this occurs might depend on the acquisition of contingency awareness during learning (Montgomery and Kirsch, 1997; Locher et al., 2017)”.

12) In both studies, pain (and relief) ratings were collected "intermittently," yet the authors make strong assumptions about effects on relief/pain based on the correlations between ratings and the time-varying measures of associability or prediction error. The authors should present complete information about rating measurement for each experiment (e.g. number of ratings) and justify why they did not incorporate ratings at the same time scale of choice, stimulus display, and SCR measurement. To determine that ratings are preferentially related to associability and not prediction error, it seems that all quantities should be measured with the same number of observations. Furthermore, this would allow direct fits to ratings, which would be the best way to determine how these learning-related parameters modulate pain and relief. Finally, if I understand correctly, Experiment 2 included pain ratings before relief outcomes were delivered. These ratings are likely to be influenced by anticipation and uncertainty, but not by relief, whereas Experiment 1's ratings were measured after outcomes. Thus, the studies differ in terms of the construct that is captured by ratings. Since pain and relief are ultimately subjective, a more thorough consideration of the self-report measures is warranted.

The ratings were collected intermittently because interruptions on each trial will impede the relief learning process, as well as greatly lengthening the duration of experiment. This is typical in pain studies. More precisely, while choices and stimulus display are inherent components of the learning paradigm and SCRs were measured without participants’ conscious input, pain/relief ratings required participants to disengage from learning and switched their attention to evaluate their perception and convert that to numerical ratings. To minimise this disruption, we decided to use intermittent ratings, which had been used previously in both appetitive and aversive conditioning paradigms (Delgado et al., 2011; Prevost et al., 2013). In addition, the tonic pain stimulation would be prolonged with ratings in each trial. Balancing the needs for participants to learn effectively, to have enough trials, and to minimise their exposure to tonic pain motivates our design. Of course, fewer ratings comes at the expense of power, but it does not invalidate the basis for model fitting and comparison, and the fact that the effects are robust reflects the effect sizes.

We now add the following sentence in the Results section:

“Ratings were taken on a sample of trials, so as to minimise disruption of task performance”

We have also included a table to summarise the details of rating timing and frequency:

See Response #4 above.

The rating timing was after outcome in Experiment 1, and before outcome in Experiment 2, which was done for precisely the reason that the reviewer mentions, that is, to address the issue that the effects might be restricted to outcome times. This issue is discussed in subsection “Summary of Experiment 1”, where we wrote:

“Second, does the modulation of pain ratings occur throughout the trial? In the task, pain ratings are taken at the outcome of the action, and only when relief is frustrated, raising the possibility that it reflects an outcome-driven response, as opposed to learning-driven process modifying the ongoing pain.”

On the issue of ratings as measures, clearly pain is necessarily subjective as the reviewer points out, but still under an endogenous control process that is objectively testable. Hence, we can test how these subjective ratings change as a result of learning (against a null hypothesis of constant pain/relief ratings since the temperature of tonic pain and reduction never changed). In other words, by formally modelling the learning process we were able to capture these subjective perceptual changes using predictors such as associability, regardless of the timing of rating collection. This also suggested the robustness of learning effects on perception. We have now added the following sentence in the Materials and methods section:

“Although it is the case that subjective ratings are inherently subjective, their modulation reflects an objective process that may explain a component of this apparent subjectivity.”

13) Please also explain the negative coefficients in Figure 4E – participants experienced LESS pain with higher associability and with longer time since relief? This seems inconsistent with previous work on uncertainty, attention, and desire for relief which should enhance pain.

Precisely, we show that participants experienced less pain with higher associability as a result of relief learning – when uncertainty is high and hence the need to learn is maximal, pain is reduced, presumably to facilitate learning. This is contrary to the effect of uncertainty seen in some experiments, which don’t involve controllable relief. This underscores the significance of the finding. As for the reduced pain with longer time since relief, it is may reflect a peripheral habituation process. We have included this as a regressor of no interest in all neuroimaging analysis to exclude its effect.

We hope that the new discussion of uncertainty in response to several points above addresses this concern.

14) Skin conductance was found to be best fit by associability from the hybrid model. Can the authors rule out the possibility that this is only the case because both associability and skin conductance decrease over time? Other models included an effect of time/trial to account for such habituation. Are these findings artefactual, and might SCR track value or prediction error if habituation is modeled separately?

Indeed, this is an issue that we considered following Experiment 1 and was part of the motivation for the non-stationary design in Experiment 2, in which the associability is maintained over time (Author response image 2). The modified discussion of this point appears in the Results section as follows:

“Third, the action-outcome contingencies were *non-stationary*, such that the relief probability from selecting each cue varied slowly throughout the experiment duration, controlled by a random walk algorithm which varied between 20-80%. This ensured that associability varied constantly through the task, encouraging continued relief exploration, and allowed us to better resolve more complex models of uncertainty. It also reduced the potential confounding correlation of associability and general habituation of SCRs.”

In addition, we have taken the precaution in model fitting procedure by (a) excluding first two trials in each session for fitting, since they are most likely to be very large due to startle effects, hence accentuating the SCR decrease, (b) constraining the variance in free parameter priors, to reduce the generalisability of associability traces so that the fitting explains mostly trial-by-trial variation (figure not included in manuscript).

**Author response image 2. respfig2:** 

15) The study uses a mild tonic stimulus in healthy volunteers and measures behavioral correlates of intermittent relief. While the pgACC results are cool, I find it quite inappropriate to suggest that "the results highlight the pgACC as a target for therapeutic intervention […] by invasive excitatory deep brain stimulation."

We have removed the suggestion of ‘invasive excitatory deep brain stimulation’ as intervention in the sentence and added ‘potential’ as a preface to ‘target’. This sentence, in the Discussion section, now reads:

“In addition to suggesting a possible computational mechanism that might underlie pain susceptibility in these patients, the results highlight the pgACC as a potential target for therapeutic intervention.”

[Editors' note: further revisions were requested prior to acceptance, as described below.]

Reviewer #1:The authors have largely satisfied any concerns I had about the reliability of the findings. So, I would be comfortable publishing the paper in its current form.That said, I concur that they haven't done as much as they might have to increase the paper's accessibility to a general audience. On re-reading the paper after the authors' responses, I think the easiest way to do this might be to do some additional revision to the introduction. The Introduction (particularly before the addition of the sections on associability and reward learning) does little to set up the actual experimental paradigms and modelling techniques used, such that one ends up trying to piece together the rationale for most of what was done while reading the methods and results. The methodology and modelling would have been far clearer to me had the authors been more explicit (as they were in their reply to reviewers) about the relevance of associability for illuminating the distinction between state and action learning and how doing so relates to the broader goal of understanding relief learning in the context of tonic pain.So, in summary, the paper is publishable, but I do think the paper could be improved in terms of accessibility without a great deal of additional work.

Thanks for this comment. We have now gone through the introduction in detail to try and better communicate the hypothesis and scientific motivation for the study. This includes:

i) justify actual experimental paradigm and modelling techniques used,

ii) discuss relevance of associability and

iii) distinction between state and action learning:

The revised Introduction:

“Tonic pain is a common physiological consequence of injury, and results in a behavioural state that favours quiescence and inactivity, prioritising energy conservation and optimising recuperation and tissue healing. This effect extends to cognition, and decreased attention is seen in a range of cognitive tasks during tonic pain (Moore et al., 2012; Crombez et al., 1997; Lorenz and Bromm, 1997). However, in some circumstances this could be counter-productive, for instance if attentional resources were required for learning some means of relief or escape from the underlying cause of the pain. A natural solution would be to suppress tonic pain when relief learning is possible. Whether and how this is achieved is not known, but it is important as it might reveal central mechanisms of endogenous analgesia. […] The studies presented here set two goals: to delineate the basic neural architecture of relief learning from tonic pain (i.e. pain escape learning) based on a state and action learning RL framework; and to understand the relationship between relief learning and endogenous pain modulation i.e. to test the hypothesis that an attentional learning signal reduces pain. We studied behavioural, physiological and neural responses during two relief learning tasks in humans, involving (i) static and (ii) dynamic cue-relief contingencies. These tasks were designed to place a high precedence on error-based learning and uncertainty, as a robust test for learning mechanisms and dynamic modulation of tonic pain. Using a computationally motivated analysis approach, we aimed to identify whether behavioural and brain responses were well described as state and/or action RL learning systems and examined whether and how they exerted control over the perceived intensity of ongoing pain.”

Reviewer #2:The authors have addressed all my comments and questions.

We thank the reviewer for the assessment.

Reviewer #3:For the most part, the authors have addressed all major concerns. I was particularly impressed that results and conclusions hold (1) whether parametric modulators are based on individual versus mean fits for Experiment 2 (although I think the authors should consider including these results in Supplementary figures), and (2) when consistently defined ROIs are employed (new Tables 6 and 7). The paper is also strengthened by the addition of information clarifying rating procedures and depicting skin conductance over time.However, I feel that a few concerns remain, which I have delineated as minor concerns in the following section. In several places (e.g. the discussions of contingency awareness, modeling the time course of temperature changes, subjectivity of ratings), I felt that the authors only superficially engaged with reviewers' collective suggestions, and that overall accessibility of the work is still somewhat limited for non-expert audiences.

We hope the revisions made in the introduction detailed above improve overall accessibility. We have addressed the reviewer’s minor comments in more detail below, but we also revisited some of the issues mentioned above:

The role of subjective awareness. We think this is important, especially for understanding pgACC function and the link to other paradigms. Although not something that we could directly address with our current design. But to highlight the importance, we now add the following comment in the Discussion section:

“However, an open question remains about the role of conscious awareness in driving pgACC-related endogenous control – a factor that is often important in these other paradigms. Whether or not the role of associability is modulated by the metacognitive awareness of uncertainty or controllability would be an important question for future studies.”

Modelling the time course of temperature. This is critical for an understanding of homeostatic motivation in which decisions would have a persistent effect on the baseline tonic temperature. Again, whilst it is beyond what we aimed to look at here, we now add further comment to emphasize that it deserves further study (Subsection “fMRI GLM model”):

“However, representation of the baseline temperature as a continuous function is clearly important in real-life contexts in which the baseline level determines homeostatic motivation and phasic reward functions (Morville et al., 2018), and hence future studies could directly manipulate this.”

On subjective ratings: we think the point at which this has the capacity to be most important is if the subjective (relief) ratings lead to a better determinant of the learned state and action values than fixed objective functions. Any effect here would be subtle if it exists at all, although it has some theoretical importance since it is against the conventional wisdom of the ‘wanting vs liking’ dissociation. Anyhow, we now add the following (subsection “Experimental Design’):

‘This does raise the issue of whether the subjective relief ratings influence the outcome values when learned in the RL model, but this (presumably subtle) effect is something that is beyond the experimental power of this experiments to resolve.”